# Light Tailoring: Impact of UV-C Irradiation on Biosynthesis, Physiognomies, and Clinical Activities of *Morus macroura*-Mediated Monometallic (Ag and ZnO) and Bimetallic (Ag–ZnO) Nanoparticles

**DOI:** 10.3390/ijms222011294

**Published:** 2021-10-19

**Authors:** Sumaira Anjum, Amna Komal Khan, Anza Qamar, Noor Fatima, Samantha Drouet, Sullivan Renouard, Jean Philippe Blondeau, Bilal Haider Abbasi, Christophe Hano

**Affiliations:** 1Department of Biotechnology, Kinnaird College for Women, 93-Jail Road, Lahore 54000, Pakistan; aaykay28@gmail.com (A.K.K.); anza.q@yahoo.com (A.Q.); noor.fatima9897@gmail.com (N.F.); 2Laboratoire de Biologie des Ligneux et des Grandes Cultures, INRAE USC1328, University of Orléans, CEDEX 2, 45067 Orléans, France; samantha.drouet@univ-orleans.fr (S.D.); hano@univ-orleans.fr (C.H.); 3Institut de Chimie et de Biologie des Membranes et des Nano-objets, CNRS UMR 5248, Bordeaux University, 33600 Pessac, France; sullivan.renouard@u-bordeaux.fr; 4Conditions Extrêmes et Matériaux: Haute Température et Irradiation (CEMHTI) CNRS UPR3079, 1D Avenue de la Recherche Scientifique, 45071 Orléans, France; jean-philippe.blondeau@univ-orleans.fr; 5Department of Biotechnology, Quaid-i-Azam University, Islamabad 15320, Pakistan; bhabbasi@qau.edu.pk

**Keywords:** UV-C irradiation, monometallic NPs, bimetallic NPs, anticancerous, anti-diabetic, anti-aging, anti-glycation, biocompatibility

## Abstract

A nano-revolution based on the green synthesis of nanomaterials could affect all areas of human life, and nanotechnology represents a propitious platform for various biomedical applications. During the synthesis of nanoparticles, various factors can control their physiognomies and clinical activities. Light is one of the major physical factors that can play an important role in tuning/refining the properties of nanoparticles. In this study, biocompatible monometallic (AgNPs and ZnONPs) and bimetallic Ag–ZnONPs (0.1/0.1 and 0.1/0.5) were synthesized under UV-C light irradiation from the leaf extract of *Morus macroura*, which possesses enriched TPC (4.238 ± 0.26 mg GAE/g DW) and TFC (1.073 ± 0.18 mg QE/g DW), as well as strong FRSA (82.39%). These green synthesized NPs were evaluated for their anti-diabetic, anti-glycation, and biocompatibility activities. Furthermore, their anti-cancerous activity against HepG2 cell lines was assessed in terms of cell viability, production of reactive oxygen/nitrogen species, mitochondrial membrane potential, and apoptotic caspase-3/7 expression and activity. Synthesized NPs were characterized by techniques including ultraviolet-visible spectroscopy, SEM, EDX, FTIR, and XRD. UV-C mediated monometallic and bimetallic NPs showed well-defined characteristic shapes with a more disperse particle distribution, definite crystalline structures, and reduced sizes as compared to their respective controls. In the case of clinical activities, the highest anti-diabetic activity (67.77 ± 3.29% against α-amylase and 35.83 ± 2.40% against α-glucosidase) and anti-glycation activity (37.68 ± 3.34% against pentosidine-like AGEs and 67.87 ± 2.99% against vesperlysine-like AGEs) was shown by UV-C mediated AgNPs. The highest biocompatibility (IC_50_ = 14.23 ± 1.68 µg/mL against brine shrimp and 2.48 ± 0.32% hemolysis of human red blood cells) was shown by UV-C mediated ZnONPs. In the case of anti-cancerous activities, the lowest viability (23.45 ± 1.40%) with enhanced ROS/NOS production led to a significant disruption of mitochondrial membrane potential and greater caspase-3/7 gene expression and activity by UV-C mediated bimetallic Ag–ZnONPs (0.1/0.5). The present work highlights the positive effects of UV-C light on physico-chemical physiognomies as well as the clinical activities of NPs.

## 1. Introduction

Nanoparticles (NPs) are nanoscale materials which have dimensions of 100 nm or less. At present, the interest of scientists in NP research is intense due to their enhanced chemical stability, thermal conductivity, nonlinear optical performance, and catalytic activity [1]. These properties make them applicable for bio-labeling, drug delivery, medicines, biosensors, optics, cosmetics, electronics, foods, and biomedicine [2]. Prior to the application of NPs their synthesis is required, and the two main conventional approaches to synthesis include physical and chemical methods. These methods are less time consuming but on the other hand lead to increased use of toxic chemicals and harsh synthesis conditions [3]. Considering the pernicious environmental and health hazards associated with conventional methods, green technology contemplates the use of plant extract-mediated biosynthesis of NPs. This approach limits the use of toxic chemicals and is eco-friendly, biocompatible, and cost effective. There are no additional impurities in NPs synthesized using the green approach; they can therefore be utilized as naive entities for biological applications [4].

Numerous plants have been exploited for the synthesis of silver (Ag), zinc oxide (ZnO), and Ag–ZnO bimetallic nanoparticles. However, use of *Morus macroura* leaf extract for the synthesis of AgNPs, ZnONPs, and Ag–ZnO bimetallic NPs has not been reported before. *M. macroura* is a tropical plant which exhibits many medicinal properties, including antimicrobial, wound healing, and anti-ulcer activity [5]. *M. macroura* is also naturally enriched with phytochemicals such as polyphenol, flavonoids, tannins, and polysaccharides, making it a suitable plant for the green synthesis of various types of NPs [6]. Plant extracts have bioactive compounds which work as a capping agent, helping in the reduction of ions to form NPs [7].

The chemical and physical characteristics of NPs can be refined by UV-C irradiation during synthesis. UV irradiation is reported to play a role in controlling the shape and size of NPs. Anjuliee et al. studied the effect of UVA and UVB light on the mobility and dissolution of silver nanoparticles. A 5-fold increase in mean diameter and 25-fold increase in Ag+ release were reported after 3 days of UV exposure [8]. Recently the UV-mediated synthesis of AuNPs and AgNPs was studied by the utilization of cornelian cherry fruit extract. Exposure to UV light of 365 nm at room temperature for 2.5 h resulted in marked changes in the size of NPs [9]. Hence, UV radiation plays a role in controlling synthesis, distribution, and size of nanoparticles.

The most widely used metallic NPs are categorized into monometallic, bimetallic, and tri-metallic NPs. Monometallic NPs exhibit the property of the constituent metal, while bimetallic NPs synthesized from two different metals are more stable structures with enriched properties. Combining different metals always enhances the inherent properties of nanomaterials [10]. Monometallic as well as bimetallic NPs have been reported to be efficient anti-cancerous agents as they induce apoptosis via ROS burst [11]. Plant mediated NPs with bioactive compounds capped over them have several benefits over chemical drugs, including efficacy at lower concentrations, target specificity, and the ability to cross the blood–brain barrier [12]. NPs also display anti-aging activity by inhibiting the formation of advanced glycation end products (AGEs) by competing with carbohydrates in binding with extra-cellular matrix proteins [13]. Therefore, NPs are emerging as cosmeceuticals and are becoming an important component of personal care products [14]. For biomedical applications, the safety of the therapeutic agent is most critical. Therefore, NPs synthesized from plants are preferable as they are biocompatible and capped with natural phytochemicals [15]. 

The role UV-C light in the green synthesis of NPs has not been well-described; however, many studies have reported on the involvement of hot electron transfers between plant phytochemicals and metal precursor ions. Generally, the plant extract-mediated synthesis of metallic/metal oxide NPs is accomplished in three main phases. During the first stage, known as the activation phase, the metal ions are reduced into atomic state and nucleated simultaneously. During this phase, upon UV-C light exposure the plant active compounds (PAC) are converted into an excited state, forming free radicals [16]. Moreover, the electrons in the valence band of metal ions are excited into the conduction band, forming hot electron holes. Following this, the metal ions are bio-reduced, and during the second phase (growth phase) they coalesce together to form NPs. A further reduction of metal ions occurs via a process known as Ostwald ripening [17]. In addition to this, the reduced metal uptake dissolves oxygen to form metal oxide NPs such as ZnONPs [18]. Lastly, during the third phase known as the termination phase, NPs acquire an energetically favorable conformation, and their final shape is determined. Additionally, PAC such as polyphenols, sterols, flavonoids, triterpenes, alcoholic compounds, alkaloids, polysaccharides, glucose, and proteins/enzymes stabilize and cap NPs [19]. In contrast to the chemical and physical methods, no external capping or reducing agents are added [20]. 

In this study the green synthesis of AgNPs, ZnONPs, and bimetallic Ag–ZnONPs was carried out using *M. macroura* leaf extract under UV-C irradiation, and their biological activities were evaluated with regard to anti-aging, anti-diabetic, and anti-cancerous properties as well as biocompatibility.

## 2. Results and Discussion

### 2.1. Phytochemical Analysis of Morus macroura

#### 2.1.1. Total Phenolic Contents of *Morus macroura*

Mulberry fruits are enriched with phenolic compounds, resulting in high antioxidant activities [21]. In our study, we measured the TPC of *M. macroura* as 4.238 ± 0.26 mg GAE/g DW. Similarly, Anwar et al. found the TPC of *M. macroura* in absolute ethanol extract to be 5.62 ± 0.22 g/100 g DW GAE [22]. Farrag et al. reported the TPC of *M. macroura* to be 33 µg/mL, expressed as mg gallic acid equivalents per gram dry weight [6]. Our findings are comparable with those in the literature, suggesting a good phenolic profile of *M. macroura*, which is involved in the bio-reduction of precursor metallic ions during NP production.

#### 2.1.2. Total Flavonoid Contents of *Morus macroura*

The carbonyl and hydroxyl groups from flavonoids can act as stabilizing and reducing agents during NPs synthesis [23]. We found the TFC of *M. macroura* to be 1.073 ± 0.18 mg QE/g DW. Other studies report the TFC of *M. macroura* to be 0.90 ± 0.06 g/100 g DW catechin equivalent [22]. Our study shows that *M. macroura* is enriched with flavonoids, therefore possessing higher antioxidant activity.

#### 2.1.3. Free Radical Scavenging Activity

FRSA is well established method to screen the antioxidant activity of plant extracts in a short time period [24]. We found the FRSA of *M. macroura* to be 82.39%, which shows a positive correlation with the enriched TPC and TFC of *M. macroura*. Previously, the FRSA of *M. macroura* was reported to be 72.42% in 80% methanol [22]. Our study shows that *M. macroura* possesses significant antioxidant potential, which reflects its ability to effectively reduce precursor metal ions during NP synthesis by donating electrons or hydrogen atoms.

### 2.2. Characterization of UV-Mediated Green Synthesized AgNPs, ZnONPs, and Bimetallic Ag–ZnONPs

#### 2.2.1. UV–Visible Spectroscopy

The free electrons in metal NPs yield a surface plasmon resonance (SPR) absorption band due to the resonance of vibration of electrons with light waves. During UV–Vis characterization the appearance of peaks shows the characteristic SPR of each type of synthesized NPs [25]. For the synthesis of AgNPs, 6 ratios (1:1, 1:2, 1:5, 1:10, 1:15, and 1:20) were firstly optimized by monitoring their SPR by UV–Vis spectroscopy, and a change in color from yellowish brown to dark brown was visually observed, as shown in Figure 1A. Initially, the 1:1 and 1:2 ratios of AgNPs did not show any significant color change and peaks, but other ratios (1:5, 1:10, 1:15, and 1:20) showed a dark brown color and peculiar peaks between 410 and 450 nm, meaning that AgNPs were synthesized in all these ratios. Among all ratios, the 1:10 ratio was selected after optimization as this ratio showed faster synthesis and earlier stabilization of AgNPs as compared to other ratios, as shown in Figure 1B. Stability was also evaluated after 1 month of incubation, and the reaction mixture showed the same value of absorbance at the same wavelength, suggesting a higher stability of synthesized AgNPs.

Using the 1:10 control, AgNPs were formed after 2 h, showing a broad peak at 450 nm as shown in Figure 2A. The high absorbance intensity reflects formation of AgNPs in higher amounts because of the reduction of the silver ions in the silver nitrate precursor salt. Moreover, the broader peak suggests the formation of larger-sized AgNPs. The UV-visible characterization of UV-C mediated AgNPs showed a sharp and narrow peak at 450 nm with greater absorbance intensity (Figure 2B). According to the literature, AgNPs display a distinctive SPR peak in the range of 400–500 nm [26]. These findings are in accordance with those of Anandalakshmi et al. [27], who reported absorption peaks of AgNPs in the range of 430–450 nm. Filip et al. [9] also reported the appearance of an SPR peak at about 418 nm for UV-light mediated green synthesis of AgNPs using cornelian cherry fruit extract. The difference in the characteristic peak and absorbance intensity between control AgNPs and UV-C mediated AgNPs suggests that UV-C light played a role in modeling the characteristics of AgNPs.

The UV-visible spectrum of green ZnONPs is depicted in Figure 2C,D. Control ZnONPs displayed a characteristic peak at 350 nm, while UV-C mediated ZnONPs also showed SPR at about 350 nm but with stronger absorption. The sharp absorption peak displayed by UV-C mediated ZnONPs could be due to a smaller particle size as compared to the control ZnONPs. The UV-visible spectra of our ZnONPs are consistent with the literature, as ZnONPs display a good absorption in the UV region of 200–400 nm [28]. The ZnONPs synthesized from *Laurus nobilis* leaf extract also show typical absorption peaks around 350 nm due to their large excitation binding energy at room temperature [29].

The UV–Vis absorption spectrum for the solutions containing bimetallic nanoparticles of Ag–ZnO (0.1/0.1) is shown in Figure 3A,B. The effect of the particles structures is witnessed in the position and intensity of localized surface plasmon resonance (LSPR). The characteristic LSPR band maximum at 400–480 nm indicates the formation of AgNPs, and the absorption peak between 350 and 380 nm depicts the synthesis of ZnONPs. In total, 2 peaks at the 350 nm and 440 nm wavelengths at the same intensity were observed for the control Ag–ZnO (0.1/0.5) bimetallic NPs. Meanwhile, UV-C mediated Ag–ZnO (0.1/0.1) bimetallic NPs displayed sharper peaks with stronger absorbance at same wavelength. Hameed et al. [30] also reported the formation of *Silybum marianum*-synthesized bimetallic Ag–ZnO NPs using UV–Vis spectroscopy. The absorption spectrum of the Ag–ZnO centered at 374 nm corresponded to ZnO, and that at 420 nm corresponded to AgNPs on the surface of zinc oxide, thereby confirming the formation of Ag–ZnO bimetallic NPs.

A broad absorption band with 2 shoulders at 350 and 430 nm was observed for control Ag–ZnO (0.1/0.5) bimetallic NPs, signifying Ag-rich and ZnO-rich bimetallic particle formation (Figure 3C,D). Moreover, UV-C mediated Ag–ZnO (0.1/0.5) bimetallic NPs displayed 2 absorption peaks at 350 and 450 nm at the same absorbance intensity. Absorption peaks were in high wavelength region, which clearly suggested the formation of bimetallic Ag–ZnO NPs. Similarly, 2 peaks for Ag/ZnO bimetallic NPs were measured between 330 nm and 366–379 nm, confirming the formation of bimetallic Ag/ZnO alloy NPs from *Moringa oleifera* leaf extract [31].

#### 2.2.2. Fourier-Transform Infrared Spectroscopy (FTIR)

FTIR analysis allowed us to identify the plant active compounds involved in the reduction, capping, and stabilization of AgNPs, ZnONPs, and bimetallic Ag–ZnONPs. FTIR analysis of *M. macroura* leaf extract showed intense peaks at ~1047.22 cm^−1^, 1623.87 cm^−1^, 2356.74 cm^−1^, and 3313.32 cm^−1^ (Figure 4). These peaks correspond to bond stretches of C=O (carbonyl group), C=C, alkyl C-H stretch, and hydroxyl groups, respectively. Bunghez et al. [32] also found an FTIR peak for mulberry extract at 3325 cm^−1^ assigned to hydroxyl groups.

FTIR spectra observed for control AgNPs displayed absorption peaks at ~1043.37 cm^−1^, 1627.73 cm^−1^, 2356.74 cm^−1^, and 3294.04 cm^−1^ (Figure 5A). Meanwhile, intense peaks at ~1047.22 cm^−1^, 1635.44 cm^−1^, 2354.81 cm^−1^, and 3375.04 cm^−1^ were observed for UV-C mediated AgNPs (Figure 5B). The peaks at 3300–3500 cm^−1^ reflect strong N-H stretching in alcohols as well as phenolic and flavonoid compounds. The peaks at 1627.73 cm^−1^ and 1635.44 cm^−1^ correspond to carbonyl stretching frequency, as reported by Some et al. at 1630 cm^−1^ for AgNPs synthesized from *Morus indica* leaf extract [33]. The FTIR spectrum of AgNPs-and *M. macroura* leaf extract revealed strong interactions with the fruit extract during their synthesis.

FTIR analysis of control ZnONPs displayed absorption peaks at 1047.22 cm^−1^, 1629.66 cm^−1^, 2343.24 cm^−1^, and 3307.54 cm^−1^ (Figure 5C). C-O and alkyl halide (C-F) strong stretching is linked with the absorption peak at 1047 cm^−1^. The peak near 1629 cm^–1–1^ is due to C=O stretching, while those at 2343 cm^−1^ and 3307 cm^−1^ are linked to the C-O bond and C-H stretching of alkenyl, respectively. The weaker band at 1629 cm^−1^ corresponds to amide-I arising due to carbonyl stretching in protein. The peaks observed for UV-C mediated ZnONPs (Figure 5D) were at 1060 cm^−1^ (C-O stretch), 1629 cm^−1^ (C=O stretching), 2366 cm^−1^ (–C=O– bond stretching of carbonyl group), and 3313 cm^−1^ (C-H stretching of alkenyl). Results mainly showed the presence of polyphenolic compounds in synthesized ZnONPs. Yedurkur et al. [34] also assigned a peak observed at 1018.12 cm^−1^ to C-O stretching in amino acids recorded for green synthesized ZnONPs. Functional groups like C-O show the involvement of polyphenols or flavonoids in ZnONP synthesis.

The FTIR spectra for control and UV-mediated 0.1/0.1 Ag–ZnO bimetallic NPs were recorded in the ranges of 1650–1800 cm^−1^, 2100–2260 cm^−1^, 2850–3100 cm^−1^, 3200–3400 cm^−1^, and 3300–3500 cm^−1^, matching C=O stretch, alkyl C-H stretch, O-H stretch, and N-H stretch, respectively (Figure 6A,B). These functional groups show the possible involvement of carboxyl acid, phenols, and amine groups in bio-reduction and stabilization of Ag–ZnONPs. Similar peaks in the range of 3300–1043 cm^−1^ with some minor differences were observed for control and UV-mediated 0.1/0.5 Ag–ZnO bimetallic NPs (Figure 6C,D). In accordance with our findings, a peak at 1630 cm^−1^ was observed for Ag-doped ZnONPs, corresponding to the stretching of the OH group [35]. Overall, in the synthesis of the abovementioned monometallic or bimetallic NPs, FTIR data showed the involvement of polyphenols, amides, and carbonyl groups as capping agents of NPs. 

#### 2.2.3. Scanning Electron Microscope Analysis

SEM analysis revealed the morphology, size, and agglomeration of green synthesized NPs. The SEM image of control AgNPs shows agglomerated irregularly shaped NPs with a size range of 54.65–51.23 nm (Figure 7A). In contrast, the SEM image of UV-C mediated AgNPs shows the formation of spherical-shaped AgNPs as well as a reduction in size 32.56–28.21 nm (Figure 7B). Rheima et al. [36] also reported the synthesis of AgNPs with hexagonal and spherical shapes and an average size of 20.23 nm under UV irradiation. In accordance with our finding, Nhunga et al. [37] reported the fast and simple synthesis of spherical-shaped AgNPs (27.9 nm) under the assistance of light. Here we can conclude that UV-C irradiation disintegrated AgNPs into a smaller size with a more defined shape of NPs as compared to the control, as reported previously [38].

SEM analysis of control ZnONPs shows the formation of agglomerated irregular rod-shaped ZnONPs with a 69.23–75.23 nm size range (Figure 7C). Likewise, the formation of agglomerated spherical-shaped phyco-mediated ZnONPs with a size range of 20–50 nm was reported in another study [39]. Moreover, UV-C irradiation promoted the synthesis of non-agglomerated, smaller-sized 57.23–62.98 nm polyhedral mostly orthorhombic cone ZnONPs (Figure 7D). Similar to our findings, Kumar et al. [40] also described the formation of hexagonal/orthorhombic-shaped ZnONPs (24.75 nm) synthesized via green route.

The variation in salt concentration and synthesis conditions influenced the morphology of bimetallic Ag–ZnONPs, as evident in the SEM images. Morphologically, the control bimetallic Ag–ZnONPs (0.1/0.1) were rod-shaped AgNPs (72.23–78.43 nm) deposited onto orthorhombic ZnONPs (56.21–59.23 nm), as evident in Figure 8A. In contrast, SEM images of UV-C mediated bimetallic Ag–ZnONPs (0.1/01) showed that AgNPs (rod-shaped) and ZnONPs (orthorhombic) had the same morphology as their respective control (Figure 8B). However, a reduction in particle size was observed in the case of UV-C irradiation, where the AgNP size was 52.13–56.23 nm and the ZnONP size was 41.51–46.44 nm. Furthermore, an increase in zinc acetate molarity refined the morphology of bimetallic Ag–ZnONPs (0.1/0.5). Control Ag–ZnONPs were flower-shaped with a particle size in a range of 43.24–33.11 nm, and we could not clearly differentiate between Ag and ZnO nanoparticles, as shown in Figure 8C. Meanwhile, UV-C irradiation promoted the formation of smaller sized bimetallic Ag–ZnONPs (0.1/0.5); values of 23.41–29.11 nm were reported for ZnO and 25.78–27.23 nm for Ag (Figure 8D). Additionally, distinct orthorhombic AgNPs and needle ZnONPs were formed, in contrast to their control in which no clear difference was observed between Ag and ZnO nanoparticles. A variety of shapes of bimetallic Ag–ZnONPs (such as spherical, plate, and rod shapes) has been reported previously [41,42]. The difference in morphology can be attributed to differences in plant extracts, synthesis conditions, and precursor salt ratios. Nonetheless, we can conclude that high-energy UV-C radiation fine-tuned the shape and size of all types of synthesized monometallic and bimetallic NPs.

#### 2.2.4. Energy Dispersive X-ray Analysis

The elemental composition and extent of purity of green synthesized NPs was evaluated by EDX. The EDX spectra of control AgNPs, as shown in Figure 9A, show the percentages of silver and oxygen to be 44.54% and 55.46%, respectively. In contrast, the EDX spectra of UV-C mediated AgNPs showed an enhancement in the weight percentage of Ag (48.77%). EDX analysis also confirmed the purity of AgNPs (Figure 9B). Strong peaks were observed at 3 keV for both control and UV-C mediated AgNPs, which are characteristic of AgNPs [43].

EDX analysis of control and UV-C mediated ZnONPs shows the purity of ZnONPs synthesized by using *M. macroura* plant extract (Figure 9C,D). Moreover, an increase in weight % of Zn from 32.68% to 41.18% was observed when ZnONPs were green synthesized under UV-C illumination. A characteristic peak of Zn was observed at 1 keV, as reported in the literature [44].

The EDX spectra of bimetallic Ag–ZnONPs at a 0.1/0.1 ratio showed the weight percentage of Zn to be 61.67% and that of Ag to be 9.62%. In contrast, bimetallic Ag–ZnONPs (0.1/0.5) with greater zinc acetate molarity showed an increase in weight % of Ag (21.22%) and Zn (50.62%). These results are in accordance with those of Sumbal et al. [41]. Moreover, UV-C illumination resulted in enhanced weight % for both types of bimetallic Ag–ZnONPs, suggesting the role of UV-C light in enhancing the bio-reduction of NPs during green synthesis. Characteristic peaks at 1 keV and 3 keV were observed for Zn and Ag, respectively. Lastly, all bimetallic Ag–ZnONPs were of high purity as their elemental composition was of O, Zn, and Ag only, as depicted in Figure 10A–D. Alharthi et al. [45] reported on the incorporation of Ag into ZnO when Ag–ZnONPs were synthesized using potato peeland and characterized by EDX.

#### 2.2.5. X-ray Diffraction Analysis

To confirm the crystalline nature of AgNPs, XRD analysis was performed. The diffraction peaks for control AgNPs with 2θ were at 31.5°, 36.2°, and 35.95°. The X-ray diffractogram pattern for control AgNPs suggests a poor crystalline nature of synthesized AgNPs. In contrast, the high-intensity peaks for UV-C mediated AgNPs that appeared with 2θ were at 38.45° (111), 44.49° (200), 65.51° (220), and 77.47° (331), as shown in Figure 11A,B. These peaks are attributed to the face-centered cubic (FCC) structure of green synthesized AgNPs, in coherence with the standard JCPDS Card NO. 04-0783. Li et al. [46] reported a similar face plane for AgNPs synthesized from corn silk aqueous extract. Clear differences in the crystallinity of AgNPs synthesized with and without UV-C illumination exist, showing the role of UV-C in refining the morphology and crystallinity of AgNPs.

The XRD pattern of control ZnONPs showed that the peaks recorded with 2θ were at 31.85°, 34.51°, 36.17°, 47.49°, 56.45°, 63.05°, and 67.85° (Figure 11C). For UV-C mediated ZnONPs, the strong peaks recorded with 2θ were at 31.99°, 34.59°, 36.21°, 47.67°, 56.85°, 62.99°, 68.15°, and 69.19° (Figure 11D). Diffraction peaks for both types of ZnONPs corresponded to the 100, 002, 101, 102, 110, 103, 200, and 201 reflection planes of the hexagonal structure of ZnONPs (JCPDS 36-1451). A similar crystalline structure of ZnONPs synthesized using *Cayratia pedata* leaf extract was reported recently [47].

XRD pattern of control bimetallic Ag–ZnO (0.1/0.1) showed greater ZnO peaks at 31.75° (100), 33.75° (002), 36.4° (101), 47.45° (102), 56.65° (110), and 62.95° (103), while UV-C mediated Ag–ZnO (0.1/0.1) displayed face plane peaks at 68.1° (112), 31.6° (100), 34.55° (002), 36.05° (101), 47.65° (102), 56.75° (110), 63.05° (103), and 68.15° (112) (Figure 12A,B). Furthermore, the control bimetallic Ag–ZnO (0.1/0.5) displayed similar peaks at 31.55°, 34.3°, 36.45°, 47.6°, 56.45°, 62.75°, and 67.85°, whereas UV-C mediated Ag–ZnO (0.1/0.5) showed peaks at 31.9°, 36°, 36.45°, 47.65°, 56.75°, 62.95°, and 68.1° (Figure 12C,D). These correspond to the 100, 002, 101, 102, 110, and 103 reflection planes of an orthorhombic nature. However, the intensities of both types of UV-C mediated Ag–ZnONPs were higher than those of their respective controls, suggesting a good alignment of atoms within the lattice plane. Similarly, the peaks of both types of UV-C mediated Ag–ZnONPs were narrower than those of their respective controls, suggesting a smaller particle size of synthesized NPs. Furthermore, both types of bimetallic NPs formed a diffractogram pattern which has a higher ZnO weight percentage as compared to silver, showing the masking effects of zinc over silver. Therefore, bimetallic Ag–ZnONPs represent a crystalline nature similar to that of monometallic NPs. As reported by Sumbul et al. [41] bimetallic Ag–ZnO NPs (0.1/0.1) also displayed major peaks of Ag, hence behaving like monometallic NPs. Moreover, the sizes of all types of synthesized monometallic and bimetallic NPs were also calculated using the Debye–Scherrer equation, as shown in Table 1. The sizes corresponded with the sizes measured by SEM, with some minor differences for all types of synthesized nanoparticles except the control AgNPs. The control AgNPs showed a greater diameter as compared to that calculated by SEM, which could be attributed to the poor crystalline nature of these nanoparticles.

### 2.3. Biocompatibility Studies of Green Synthesized AgNPs, ZnONPs, and Bimetallic Ag–ZnONPs

The brine shrimp lethality assay was developed as a rapid method to predict the toxic effects of NPs [48]. In our study, all NPs were found to be moderately toxic towards brine shrimp, as the LC_50_ value was between 10.0 and 30.0 µg/mL, as shown in Figure 13A [49]. UV-C mediated AgNPs were found to have a moderately toxic LC_50_ (22.13 ± 2.91 µg/mL) as compared to control AgNPs, which were slightly less toxic (LC_50_ 14.77 ± 2.91 µg/mL). Recently, the low toxic nature of chitosan-loaded green synthesized AgNPs was reported, where an LC_50_ = 518.41 µg/mL was considered safe [50]. Control ZnONPs were also moderately toxic with an LC_50_ of 19.23 ± 1.45 µg/mL, while UV-C mediated ZnONPs were less toxic (LC_50_ 14.23 ± 1.68 µg/mL). Similarly, Paul et al. [51] demonstrated the bio-safe nature of *Coriander oleoresin*-mediated ZnONPs towards brine shrimp. 

Control bimetallic Ag–ZnONPs (0.1/0.1 and 0.1/0.5) showed similar toxicity towards *Artemia* with LC_50_ values of 21.57 ± 4.24 µg/mL and 21.23 ± 1.10 µg/mL, respectively. UV-C mediated bimetallic Ag–ZnONPs (0.1/0.1) were moderately toxic with an LC_50_ of 20.50 ± 2.36 µg/mL. UV-C mediated bimetallic Ag–ZnONPs (0.1/0.5) showed an LC_50_ of 19.80 µg/mL. Hameed et al. [30] reported a decrease in the cytotoxicity of green synthesized bimetallic Ag–ZnONPs with reduced concentrations of NPs. It was also observed that the LC_50_ value of bimetallic NPs tended towards mild toxicity as compared to their mono-metallic counterparts, suggesting a more biocompatible nature of bimetallic NPs. Lastly, a trend was observed in which NPs green synthesized under UV-C illumination were slightly more moderately toxic as compared to their control, with the exception of AgNPs. This could be due to the smaller size and unique shapes of UV-C mediated NPs.

The biocompatibility of green synthesized NPs with hRBCs was evaluated in terms of % hemolysis. All NPs were found to be slightly hemolytic, as their % hemolysis values were between 2% and 5% based on the standards of the American Society for Testing and Materials Designation (Figure 13B) [52]. Control AgNPs showed 3.05 ± 0.51% hemolysis, as compared to UV-C mediated AgNPs which showed a less hemolytic nature (2.85 ± 0.22%). Similarly, the biocompatible nature of pomegranate fruit pericarp-mediated AgNPs with human RBCs was recently demonstrated by Govindappa et al. [53]. Control ZnONPs and UV-C mediated ZnONPs were also found to be slightly hemolytic towards hRBCs, as they showed 3.45 ± 0.60% and 2.48 ± 0.32% hemolysis, respectively. Likewise, Kumar et al. [52] also reported 2.7–6.4% hemolysis of RBCs when treated with 100 and 500 µg/mL of *S. glauca*-mediated ZnONPs.

Control and UV-C mediated bimetallic Ag–ZnONPs (0.1/0.1) were 2.75 ± 0.21% hemolytic and 2.97 ± 0.30% hemolytic towards hRBCs, respectively. On the other hand, control and UV-C mediated bimetallic Ag–ZnONPs (0.1/0.5) were slightly more hemolytic, with values of 3.43 ± 0.24% and 3.60 ± 0.44%, respectively. Zare et al. also confirmed the slightly hemolytic nature of Ag–ZnO nanocomposites at lower concentrations; however, they showed greater hemolysis as compared to their monometallic counterparts [54]. Here we can conclude on the bio-safe nature of green synthesized NPs, as the hemolysis of hRBCs by all types of synthesized monometallic and bimetallic NPs fell into the low range at 2–5%.

### 2.4. Anti-Diabetic and Anti-Glycation Activities of Green Synthesized AgNPs, ZnONPs, and Bimetallic Ag–ZnONPs

One way to control blood glucose level is to delay the breakdown of carbohydrates in small intestine by blocking the action of α-glucosidase and α-amylase [55]. In our study, UV-C mediated AgNPs showed significant inhibition of α-amylase (67.77 ± 3.33%) as well as α-glucosidase (35.83 ± 2.40%) as compared to control AgNPs (35.57 ± 1.97% and 20.10 ± 2.95%, respectively) (Figure 14A). The enhanced anti-diabetic activity of UV-C mediated AgNPs could be attributed to their truncated triangular morphology which successfully entraps amylase enzymes [56]. Control ZnONPs displayed 65.30 ± 2.77 % inhibition of α-amylase and 36.43 ± 3.10% inhibition of α-glucosidase. UV-C mediated ZnONPs showed lesser inhibition of carbohydrate digesting enzymes as compared to control. Arvanag et al. [57] also confirmed the outstanding anti-diabetic activity of green synthesized ZnONPs.

Control bimetallic Ag–ZnONPs (0.1/0.1) displayed better inhibitory activity (24.93 ± 1.74% α-glucosidase and 49 ± 3.98% α-amylase inhibition) as compared to control bimetallic Ag–ZnONPs (0.1/0.5) (23.27 ± 1.80% α-glucosidase and 40.6 ± 2.26% α-amylase inhibition). However, their UV-C mediated counterparts showed slightly less inhibition. Bakur et al. [58] reported similar level of anti-diabetic activities of biogenic bimetallic Ag–ZnONPs in a concentration-dependent manner. Our study shows the promise of green NPs as an effective drug in managing diabetes (Figure 14A).

Our study highlights the plausible role of UV-C light in enhancing the biological activity of NPs by refining their morphology, as UV-C mediated AgNPs showed the most significant anti-diabetic activity among all NPs.

Oxidative stress may lead to the build-up of advanced glycation end products (AGEs) [59]. Therefore, the inhibition of vesperlysine-like AGEs and pentosidine-like AGEs by NPs is emerging as an effective anti-aging tool [14]. In our study, UV-C mediated AgNPs displayed the highest inhibition of vesperlysine-like AGEs (67.87 ± 2.99%) and pentosidine-like AGEs (37.68 ± 3,70%). Control AgNPs showed 36.53 ± 3.49% and 19.78 ± 1.46% inhibition of vesperlysine-like AGEs and pentosidine-like AGEs, respectively (Figure 14B). These values are significantly higher than those of previous reports [59]. The anti-glycation activity of ZnONPs has also been reported to counter oxidative stress in neurodegenerative disease [60]. Here we report on the exceptional anti-glycation activity of green synthesized ZnONPs as anti-aging agents. Control ZnONPs showed 66.47 ± 2.67% inhibition of vesperlysine-like AGEs and 36.31 ± 1.23% pentosidine-like AGEs, while UV-C mediated ZnONPs showed 62.27 ± 0.67% inhibition of vesperlysine-like AGEs and 32.36 ± 2.86% pentosidine-like AGEs. Bimetallic Ag–ZnONPs displayed good anti-aging activity, though less than that of monometallic Ag and ZnO NPs (Figure 14B).

We found out that UV-C mediated AgNPs inhibited AGE production most significantly amongst all NPs, therefore showing the positive role of UV-C light in tailoring the morphology as well as the subsequent anti-glycation activity of NPs.

### 2.5. Anti-Cancerous Activities of Green Synthesized AgNPs, ZnONPs, and Bimetallic Ag–ZnONPs

#### 2.5.1. Cell Viability Assay by MTT

The anti-cancerous activity of NPs was evaluated in terms of % cell viability of HepG2 cells. Our results clearly demonstrate the anti-cancerous potential of green synthesized NPs as compared to non-treated cells which were 100% viable (Figure 15A). Cells treated with UV-C mediated bimetallic Ag–ZnONPs (0.1/0.1) were the least viable (23.43 ± 1.40%), while the control showed 37.86 ± 2.89% viability. Similarly, UV-C mediated bimetallic Ag–ZnONPs (0.1/0.5) were 28.9 ± 1.19% viable, while the control resulted in 34.37 ± 2.62% viability. This shows the positive role of UV-C illumination in refining morphology of NPs, hence increasing their therapeutic efficacy. Zgura et al. also reported a reduction in the cell viability of human fibroblast BJ cells when exposed to clove and mandarin peel extract-mediated Ag–ZnO nanocomposites [61].

Control AgNPs reduced the cell viability of HepG2 cells up to 36.25 ± 1.42% cell viability, while UV-C mediated AgNPs showed 31.8 ± 1.31% cell viability. Recently, a 40% death rate was reported for MCF-7 cells when treated with AgNPs [62]. Control ZnONPs resulted in 37.16 ± 1.23% viability, while UV-C mediated ZnONPs resulted in 40.92 ± 2.52% viability. These findings are in coherence with previous reports [63]. From these results we can confirm the pronounced effects of UV-C mediated NPs on reducing cell viability.

#### 2.5.2. Measurement of Intracellular ROS/RNS Production

In this study, the production of ROS/RNS was correlated with cell viability, as oxidative burst results in reduced cell viability and proliferation [64]. In NTC, 835 relative DHR123 fluorescence units were recorded, and NP treatment significantly enhanced intracellular ROS production (Figure 15B). UV-C mediated bimetallic Ag–ZnONPs (0.1/0.1) led to the production of 3441.5 ± 56.55 relative DHR123 fluorescence ROS units, while the control showed 3234.9 ± 95.62 relative DHR123 fluorescence units. Similarly, UV-C mediated bimetallic Ag–ZnONPs (0.1/0.5) produced 3487.1 ± 357.12 relative DHR123 fluorescence units, as compared to the control, which produced 2485.5 ± 287.12 relative DHR123 fluorescence units. The induction of high amounts of ROS by Ag–ZnO nanocomposites overwhelmed the anti-oxidant capacity of cells, eventually leading to the death of cancerous cells [65].

Control AgNPs also induced the oxidation of HepG2 cells, resulting in 2831 ± 801.12 relative DHR123 fluorescence units while UV-C mediated AgNPs resulted in the production of 2605.07 ± 190.43 relative DHR123 fluorescence ROS units. Chairuangkitti et al. [66] reported the ROS-dependent cytotoxicity in A549 adenocarcinoma cells by AgNPs. On the other hand, control ZnONPs led to 2567.4 ± 163.12 relative DHR123 fluorescence units. There was an increase in ROS levels when treated with UV-C mediated ZnONPs (2632.9 ± 234.12 relative DHR123 fluorescence units). The overgeneration of ROS by the liberation of Zn^2+^ ions from ZnONPs resulted in mitochondrial dysfunction in PC_12_ cells [67]. Amongst all NPs, UV-C mediated bimetallic Ag–ZnONPs (0.1/0.5) produced the highest level of intracellular ROS/RNS, showing the enhanced properties of bimetallic alloys.

#### 2.5.3. Measurement of Mitochondrial Membrane Potential

A loss of MMP with an increase in intracellular ROS levels leads to cell death [68]. Figure 15C shows a significant loss of MMP in response to control ZnONPs (2224.55 ± 86.02 RFU) as compared to NTC (3374.93 ± 105.12 RFU). Moreover, UV-C mediated ZnONPs also reduced the MMP of HepG2 cells (2058.8 RFU). Likewise, Li et al. [69] showed a collapse of MMP in response to rod-shaped ZnONPs. Control AgNPs showed 2128.5 RFU loss of MMP while UV-C mediated AgNPs resulted in a 2026.2 ± 66.77 RFU loss of MMP. In support of our results, Xue et al. [70] reported a commendable loss of MMP in HepG2 cells when treated with nano-silver of 23.44 nm in size.

As compared to their monometallic counterparts, bimetallic NPs significantly reduced MMP in HepG2 cells, particularly UV-C mediated bimetallic Ag–ZnONPs (0.1/0.1), with an MMP of 1777.17 ± 40.12 RFU. In contrast, control bimetallic Ag–ZnONPs (0.1/0.1) resulted in an MMP of 1963.3 ± 71.21 RFU. Similarly, UV-C mediated bimetallic Ag–ZnONPs (0.1/0.5) showed an MMP of 1989.20 ± 80.76 RFU, whereas the control bimetallic Ag–ZnONPs (0.1/0.5) led to an MMP of 2140.76 ± 68.21 RFU. These findings show that UV-C mediated green synthesized NPs can alter mitochondrial dynamics in cancerous cells, especially UV-C mediated bimetallic NPs which show enhanced properties.

#### 2.5.4. Caspase-3 Gene Expression and Caspase-3/7 Activity

The induction of caspase proteins increases ROS generation and mitochondrial dysfunction, causing the apoptosis of cancerous cells [71]. Exposure of HepG2 cells to control AgNPs caused an increase in caspase-3 gene expression (170.23 ± 10.02 log 2-fold change) as compared to NTC (100 ± 1.80 log 2-fold change) (Figure 16A). UV-C mediated AgNPs further enhanced the caspase-3 gene expression to 178.46 + 10.78 log 2-fold change. In addition to improving caspase-3 gene expression, control AgNPs enhanced caspase-3/7 activity to 232.2 ± 13.35 RFU/mg protein, as compared to NTC 100 ± 6.6 RFU/mg protein (Figure 16B). UV-C mediated AgNPs resulted in 255.4 ± 15.45 RFU/mg protein activity. Kim et al. also reported on the therapeutic potential of *Eleutherococcus senticosus* AgNPs against lung cancer cells by activating the expression of the caspase-3 gene [72]. In agreement with our results, *Albizia adianthifolia* mediated AgNPs increased the activity of effector caspase-3/7 proteins by 1.7-fold [73].

Similarly, control ZnONPs enhanced caspase-3 gene expression by 154.7 ± 11.07 log 2-fold change, while UV-C mediated ZnONPs further enhanced gene expression by 174.7 ± 17.03 log 2-fold change. Control ZnONPs induced 221.5 ± 15.23 RFU/mg protein caspase-3/7 activity, while UV-C mediated ZnONPs further enhanced this by 250 ± 25.12 RFU/mg protein (Figure 16A,B). ZnONPs also induced apoptosis in gingival squamous cell carcinoma via the expression of the effector caspase-3 gene [74]. Song et al. confirmed elevated caspase-3/7 activity in a primary neocortical astrocyte culture upon exposure to 67 nm ZnONPs [75].

UV-C mediated bimetallic Ag–ZnONPs (0.1/0.1) significantly increased caspase-3 gene expression with a 395.95 ± 49.47 log 2-fold change, as compared to control bimetallic Ag–ZnONPs (0.1/0.1), which showed a 248.13 ± 23.32 log 2-fold change. Likewise, control bimetallic Ag–ZnONP (0.1/0.1) elevated caspase-3/7 activity by 288.5 ± 25.12 RFU/mg protein, and its UV-C counterpart further elevated this up to 333 ± 20.57 RFU/mg protein. UV-C mediated bimetallic Ag–ZnONPs (0.1/0.5) also resulted in similar caspase-3 gene expression with a 323.16 ± 53.23 log 2-fold change, while the control resulted in a 273.26 ± 54.23 log 2-fold change. An increase in the ZnO ratio in bimetallic Ag–ZnONPs (0.1/0.5) significantly increased caspase-3/7 activity (324.4 ± 31.12 RFU/mg protein). UV-C bimetallic Ag–ZnONPs (0.1/0.5) further heightened caspase-3/7 activity to 362.5 ± 23.45 RFU/mg protein. From these findings, it can be concluded that bimetallic NPs enhanced caspase-3 gene expression and caspase-3/7 activity considerably more than monometallic NPs. Moreover, UV-C mediated NPs further enhanced caspase-3 gene expression and caspase-3/7 activity.

Nanoparticles play a role in bringing about programmed cell death by regulating Bcl_2_, P53, and caspase expression, through which the intrinsic or extrinsic pathway is initiated. These events lead to DNA fragmentation, cell bleeding, and ultimately cell death [76]. The extrinsic pathway of apoptosis starts through the binding of ligands to death receptors such as the Fas receptor on the cell surface of tumor receptors [77]. These receptors contain death domains (FADD) which possess different upstream procaspases such as caspase 8 and 10. These caspases induce the death signaling complex, which further activates caspase 3, 6, and 7. Here, caspase 8 can cause cell death in both direct and indirect ways. In the former, caspase 3 and caspase 7 are activated, whereas in latter, BH3 proteins are activated, converting BID proteins into tBID. At this point, the indirect process is linked to the intrinsic cycle [78]. The intrinsic pathway is initiated due to the induction of cellular stress ROS by stimuli like NPs. In response to these stresses, pores are formed in the mitochondrial membrane, initiating the caspase cascade. In the intrinsic apoptosis pathway, firstly cytochrome c is discharged into cytosol. Cytochrome c and APAF_1_ permeabilize the mitochondrial outer membrane, which discharges all soluble proteins and Bcl_2_ proteins to form the apoptosome [79]. The apoptosome activates caspase-9, which activates the effector caspase-3, resulting in DNA fragmentation, cell blebbing, and apoptosis of cancer cells [80].

## 3. Materials and Methods

### 3.1. Preparation of Aqueous Leaf Extract of Morus macroura

For the preparation of aqueous leaf extract, fresh *Morus macroura* leaves were collected from Botanical Garden of Kinnaird College for Women, Lahore, Pakistan, and verified by Dr. Samina Hassan (Assistant Professor, Department of Botany, Kinnaird College for Women Lahore, Pakistan). The 10 g leaves were washed first with tap water 3 to 4 times and then with distilled water thoroughly to remove dust particles, and then the leaves were shade dried. Leaves were grinded in powder form with the help of an electrical grinder and soaked in 400 mL deionized water. The mixture was heated right after 24 h at 40 ± 5 °C overnight and stirred constantly. When the remaining volume reached 100 mL the mixture was filtered with Whatman’s filter paper and stored at 4 °C for NP synthesis.

### 3.2. Phytochemical Analysis of Morus macroura

#### 3.2.1. Total Phenolic Contents

The Folin–Ciocalteu method was used to evaluate the total phenolic content (TPC) of *M. macroura* leaf extract as described by Kamtekar et al. [81]. Firstly, 1 mL of plant extract or gallic acid standard (1000, 750, 500, 250, 125, 50 μg mL^−1^) was taken in a test-tube, to which 5 mL distilled water and 0.5 mL Folin–Ciocalteu’s reagent were added and shaken well. After 5 min incubation at RT, 1.5 mL of 20% sodium carbonate was added and the total volume was increased up to 5 mL using distilled water. A deep blue color was observed. After 2 h of incubation at room temperature (RT), the calibration curve was plotted by measuring absorbance of known gallic acid concentrations at 750 nm using a spectrophotometer (Analytik Jena, Specord 200 plus, Jena, Germany). The results for TPC were measured from a gallic acid calibration curve (y = 0.00004x + 0.0089, R² = 0.9953) and expressed as mg GAE/g DW. TFC was calculated by using formula:(1)$$\mathrm{TFC}\;\left(\mathrm{GAE}\right)\;=\frac{C\times V\;\left(mL\right)}{m\;\left(g\right)}$$
where *C* = concentration of extract, *V* = volume of extract, and *m* = mass of extract. The TPC was expressed as mg/g of gallic acid equivalents in milligrams per gram (mg GAE/g) of dry extract. TPC was assessed in triplicate.

#### 3.2.2. Total Flavonoid Contents

The aluminum chloride colorimetric method was adopted to measure total flavonoid contents (TFC), as reported by Aryal et al. [82]. Briefly, 1 mL of leaf extract or quercetin standard (25–200 μg mL^−1^) was placed in a test-tube, to which 0.2 mL of 1 M potassium acetate, 0.2 mL of 10% (*w*/*v*) AlCl_3_ solution, and 5.6 mL distilled water were added, mixed well, and incubated at RT for 30 min. The calibration curve was plotted by measuring the absorbance of known quercetin concentrations at 415 nm using a spectrophotometer (Analytik Jena, Specord 200 plus). The results for TFC were measured from a quercetin calibration curve (y = 0.0057x + 0.0127, R^2^ = 0.9973) of quercetin (25–200 µg/mL) and expressed as quercetin equivalent (QE) per gram DW. TFC was calculated by using the formula:(2)$$\mathrm{TPC}\;\left(\mathrm{QE}\right)\;=\frac{C\times V\;\left(mL\right)}{m\;\left(g\right)}$$
where *C* = concentration of extract, *V* = volume of extract, and *m* = mass of extract. All the experimental work was performed in triplicate.

#### 3.2.3. Free Radical Scavenging Activity (FRSA)

The antioxidant activity of *M. macroura* leaf extract was measured by using DPPH (2, 2-diphenyl-1-picrylhydrazyl) as described by Anjum et al. [83]. In short, 0.5 mL of leaf extract and 4.5 mL DPPH (3.2 mg/100 mL methanol) were mixed and incubated at RT for 1 h. Lastly, absorbance at 517 nm was measured by using a spectrophotometer. All the experiments were run in triplicate. The FRSA was measured as percentage of discoloration of DPPH using the equation stated below:(3)$$\mathrm{FRSA}\;\left(\%\right)\;=100\times \;\left(1-\frac{A_{c}}{A_{s}}\right)$$
where A_c_ = absorbance of plant extract and DPPH and A_s_ = absorbance of DPPH solution (standard).

### 3.3. UV-Mediated Green Synthesis of Nanoparticles

#### 3.3.1. AgNPs

UV-mediated green synthesis of monometallic AgNPs was carried out as reported by Anjum et al. [84]. For this, 1 mL leaf extract and 0.01 M sliver nitrate solution were mixed in 6 ratios *v*/*v* (i.e., 1:1, 1:2, 1:5, 1:10, 1:15, and 1:20) to identify an appropriate concentration of silver nitrate for the optimal synthesis of AgNPs. The ratio of 1:10 was chosen as there was an instant color change from greenish yellow to dark brown, and characteristic peaks of AgNPs were observed when absorbance was measured between 380 and 460 nm using a UV-visible spectrophotometer. After mixing of the plant extract and precursor salt solution, it was constantly stirred at room temperature with and without UV-C light (200–280 nm) exposure. After 2 h, the NPs solution was centrifuged at 13,000× *g* rpm for 10 min. Dark brown pellets were obtained consisting of silver nanoparticles. The supernatant was discarded and then the pellets were washed with deionized water thrice. After washing, AgNPs were dried at room temperature and collected for characterization and biological applications.

#### 3.3.2. ZnONPs

Control ZnONPs were synthesized via green route following the protocol of Anjum et al. [85] with minor changes. For this, 50 mL of 0.02 M zinc acetate solution and 1 mL of leaf extract of *M. macroura* were mixed, while 2 M NaOH was added dropwise until the pH was 12. The mixture was constantly stirred using a magnetic stirrer for 2 h at RT under the illumination of a UV-C (200–280 nm) lamp and without the lamp as a control. The color change from colorless to yellowish white indicated the initiation of ZnONP synthesis. After 2 h, the ZnONP solution was micro-centrifuged at 6000× *g* rpm for 15 min, the supernatant was discarded, and the pellets were washed by re-suspending in distilled water. This step was performed thrice. Pellets of ZnONPs were dried at 40 °C overnight in an oven. Finally, the dried ZnONPs were crushed to fine powder using a mortar and pestle.

#### 3.3.3. Bimetallic Ag–ZnONPs

Bimetallic Ag–ZnONPs were prepared by following the method mentioned by Sumbal et al. [41] after minor modifications. Thus, 2 different ratios, i.e., Ag–ZnO 0.1/0.5 and Ag–ZnO 0.1/0.1, were prepared. For the synthesis of Ag–ZnO 0.1/0.5 bimetallic NPs, 1 mL of leaf extract was heated at 60 °C for 5 min and then 50 mL of 0.5M zinc acetate solution was added. The solution was placed for 5 min in the shaking incubator at 60 °C. After 5 min 10 mL of 0.1 M silver nitrate was added. Bimetallic Ag–ZnO (0.1/0.1) was prepared by the same method but with 0.1 M 50 mL zinc acetate solution. The pH of the solution was adjusted to 12 by adding 2 M NaOH. Then, the solution was kept in shaking incubator at constant stirring for 2 h at 60 °C. The solution was centrifuged at 13,000× *g* rpm for 15 min and pellets were washed with deionized water thrice. In the case of Ag–ZnO 0.1/0.5 bimetallic NPs, pellets were of greyish white color. Meanwhile, Ag–ZnO 0.1/0.1 bimetallic NPs were of a greyish black shade. Lastly, the green synthesized bimetallic NPs were air dried.

### 3.4. Characterization of UV-Mediated Green Synthesized NPs

#### 3.4.1. UV–Visible Spectroscopy

The progress of green synthesis of NPs was monitored by using UV-visible spectroscopy after a 30 min interval during a 2 h reaction. The UV–Vis spectra of the AgNP, ZnONP, and bimetallic Ag–ZnONP reaction mixtures were recorded using a spectrophotometer (Analytik Jena, Specord 200 plus, Jena, Germany) in the ranges of 380–460 nm, 300–800 nm, and at both wavelengths, respectively.

#### 3.4.2. Attenuated Total Reflection Fourier-Transform Infrared Spectroscopy (ATR-FT-IR) 

Fourier-transform infrared (FT-IR) analysis of NPs was conducted as reported by Tungmunnithum et al. [86]. A reflectance spectrum was acquired by a Brucker (Palaiseau, France) V70 interferometer working in a reflectivity mode which had an ATR accessory comprising gold crystal. The wave numbers were measured in the 400–4500 cm^−1^ wavenumber range. The resolution of the instrument was about 4 cm^−1^. The FTIR data in this study are reported as the mean of triplicates.

#### 3.4.3. Scanning Electron Microscopy (SEM) and Energy Dispersive X-ray (EDX) Analyses

The morphology of NPs was evaluated by SEM using the SIGMA model (MIRA3; TESCAN, Brno, Czech Republic), which operated at an accelerating voltage of 10 kV. Small amounts of samples were dropped on a carbon-coated copper grid. The film was dried for 5 min under a mercury lamp, and SEM images of monometallic and bimetallic NPs at different magnifications were collected. EDX analysis was performed by an EDX detector attached to an SEM for the elemental analysis of ZnONPs.

#### 3.4.4. X-ray Diffraction Analysis

The crystalline nature of NPs was investigated using X-ray diffraction (XRD). Briefly, monometallic and bimetallic NPs were coated on an XRD grid and measurements were taken in the scanning mode using an X-ray diffractometer (Shimadzu-Model, XRD6000, Nettetal, Germany) which operated at 40 kV with a current of 30 mA and Cu/kα radiation in the range of 20–80° in 2θ angles. Moreover, the sizes of all types of synthesized NPs were calculated using the Debye–Scherrer equation [29]
(4)$$D=\frac{k\lambda }{\beta \;\cos\;\theta }$$
where k = shape factor (0.94); β = full width at half maximum (FWHM) in radians; λ = X-ray wavelength (λ = 1.5418 Å); and θ = Bragg’s angle.

### 3.5. Biocompatibility Studies

#### 3.5.1. Brine Shrimp Lethality Assay

The lethality of green synthesized AgNPs, ZnONPs, and bimetallic Ag–ZnONPs (20 mg/mL stock in water) against *Artemia salina* (brine shrimp) was evaluated in a 96-well plate (300 µL) for 24 h. The larvae of brine shrimps were used for toxicological study by following protocol reported by Ahmed et al. [87]. Briefly, the eggs of *A. salina* were placed in a specifically designed plastic tray with 2 compartments containing sterile sea water (38 g/L) supplemented with 6 mg/L dried yeast. The eggs were hatched by incubation for a period of 24–48 h, with a constant supply of oxygen. Proper illumination was ensured to maintain the temperature (30–32 °C) and light necessary for hatching. Then, 10 mature (phototropic) nauplii were picked up with a Pasteur pipette and placed into the wells, to which AgNPs, ZnONPs, and bimetallic Ag–ZnONPs (25, 50, 100, and 200 µg/mL) were separately added and the final volume was adjusted to 300 µL. The negative control was 1% DMSO in sea water, while doxorubicin (1–10 µg/mL) served as a positive control. After 24 h of incubation the live shrimps were quantified and the median lethal concentration (LC_50_) was measured by using table curve 2D v5.01 of the test extracts with ≥50% mortality.

#### 3.5.2. Biocompatibility with Human Red Blood Cells (hRBCs)

The biocompatibility of AgNPs, ZnONPs, and bimetallic Ag–ZnONPs was evaluated against freshly isolated hRBCs. A blood sample from a healthy donor (male, 43 years of age) was withdrawn using a sterile syringe at the fingertip. Procedures dealing with human subjects were carried out in consideration of the ethical standards of international and national research committees and the 1964 Helsinki Declaration and its later amendments. Written informed consent was obtained from the healthy donor participating in this study. Blood was collected in tubes containing EDTA to prevent blood clotting [88]. RBCs were extracted by centrifuging 1 mL blood for 5 min at 14,000× *g* rpm. Furthermore, 200 µL of pelleted erythrocytes were shaken in 9.8 mL of PBS (pH: 7.2). In a 1.5 mL Eppendorf tube, 100 µL of AgNPs/ZnONPs/Ag–ZnONPs and erythrocytes were taken separately, incubated at 35 °C for 1 h, and centrifuged for 10 min at 10,000× *g* rpm. In a 96-well plate, 100 µL of supernatant was added and the hemoglobin released was measured at 540 nm using a BioTek ELX800 Absorbance Microplate Reader (BioTek Instruments, Colmar, France). Triton X-100 and DMSO served as a positive and negative control, respectively. The results were measured as % hemolysis using the following formula:(5)$$\%\;\mathrm{Hemolysis}=\frac{\mathrm{Abs}\;\mathrm{of}\;\mathrm{Sample}-\mathrm{Abs}\;\mathrm{of}\;\mathrm{Negative}\;\mathrm{Control}}{\mathrm{Abs}\;\mathrm{of}\;\mathrm{Positive}\;\mathrm{Control}-\mathrm{Abs}\;\mathrm{of}\;\mathrm{Negative}\;\mathrm{Control}}\times 100$$

### 3.6. Anti-Diabetic Activities of Green Synthesized NPs

#### 3.6.1. α-Glucosidase Inhibition

α-Glucosidase was purified and immobilized by using rat intestinal acetone powder (Sigma, St. Quentin Fallavier, France) and CNBr-activated sepharose 4B (Sigma, St. Quentin Fallavier, France), respectively, according to the previously described protocol of Hano et al. [89]. As such, 1 μL of intestinal fluid supplemented with 4-nitrophenyl-α-d-glucopyranoside (5 mM, 4NPG; Sigma, St. Quentin Fallavier, France) was used to perform this assay. After 30 min incubation, 1 M sodium carbonate was added, and the reaction stopped using polyethylene filter (0.45 μm) with an end-capped column. The chromogenic method was employed to evaluate the activity of immobilized enzyme against blank solution by measuring absorbance at 405 nm. The % inhibition of α-glucosidase was measured by calculating the difference in absorbance with and without the sample.

#### 3.6.2. α-Amylase Inhibition

Inhibition of α-amylase activity by NPs was measured by using α-amylase from porcine pancreas (Sigma Aldrich, St. Louis, MO, USA) according to the protocol established by Velioglu et al. [90]. Briefly, 1 μ/mL enzyme was prepared in 4-nitrophenyl-α-d-maltopentaoside (5 mM, 4NPM; Sigma) and phosphate buffer (0.1 M; pH 6.8) by constant mixing. The reaction mixture was incubated for 30 min with and without samples at RT. Then the reaction was stopped by using 1 M sodium carbonate solution followed by measuring enzyme activity by taking absorbance at 405 nm by Synergy II reader (BioTek Instruments, Colmar, France). The % inhibition of α-amylase was measured by calculating the difference in absorbance with and without the sample.

### 3.7. Vesperlysine and Pentosidine-like AGEs Activity

The anti-aging activity of monometallic and bimetallic NPs was evaluated by assessing the inhibition of production of advanced glycation end-products (AGEs) based on the protocol reported by Shah et al. [59]. The test sample was mixed with 20 mg/mL BSA solution (Sigma Aldrich) prepared in phosphate buffer (0.1 M; pH 7.4), glucose solution (0.5 M; Sigma Aldrich) prepared in phosphate buffer, and 1 mL of phosphate buffer (0.1 M; pH 7.4) containing sodium azide 0.02% (*w*/*v*). The mixture was incubated for 5 days in the dark at 37 °C, and the fluorescent AGE formation was measured using a VersaFluor fluorometer (Bio-Rad, Marnes-la-Coquette, France) set with 410 nm emission and 330 nm excitation wavelengths, respectively. The AGE inhibition was expressed as the % inhibition relative to the control.

### 3.8. Anti-Cancerous Activity of Green Synthesized NPs

#### 3.8.1. Cell Viability Assay by MTT

Human hepato-cellular carcinoma cells (HepG2) (ATCC HB-8065; American Type Culture Collection, Manassas, VA, USA) were cultured in Dulbecco’s Modified Eagle Medium for anti-cancerous studies. MTT (3-(4,5-dimethylthiazolyl-2)-2,5-diphenyltetrazolium bromide) dye was utilized to assess the cytotoxicity of AgNPs, ZnONPs, and bimetallic Ag–ZnONPs in vitro. The reduction of tetrazolium dye MTT by living cells into an insoluble purple colored product formazan was recorded using a spectrophotometer as a measure of cell viability.

Thus, 200 µg/mL of green synthesized AgNPs, ZnONPs, and bimetallic Ag–ZnONPs were added to a 96-well plate which was pre-seeded with HepG2 cells (>90% viability; 1 × 10^4^ cells/well; 200 µL per well), for 24 h. Then, 10 µL of MTT dye (5 mg/mL) was added per well and incubated for 3 h. Then, 10% acidified sodium dodecyl sulfate was added to dissolve insoluble formazan. Lastly, after the overnight incubation of cells, plates were analyzed at 570 nm using a microplate reader (Platos R, 496. AMP, AMEDA Labordiagnostik GmbH, Graz, Austria). Non-treated cells (NTC) acted as a control. The experiment was performed in triplicate. Cell viability was measured in terms of the percentage with respect to the control using the equation below:(6)$$\mathrm{Viability}\;(\%)=\frac{\mathrm{Ab}\;\mathrm{of}\;\mathrm{Sample}-\mathrm{Ab}\;\mathrm{of}\;\mathrm{Control}}{\mathrm{Ab}\;\mathrm{of}\;\mathrm{NTC}-\mathrm{Ab}\;\mathrm{of}\;\mathrm{Media}}\times 100$$

#### 3.8.2. Measurement of Intracellular Reactive Oxygen and Nitrogen Species (ROS/RNS)

The intracellular ROS/RNS level was assessed by using dihydrorhodamine-123 (DHR-123) fluorescent dye (Sigma-Aldrich, St. Quentin Fallavier, France) as reported by Nazir et al. [91]. Pre-seeded HepG2 cells in a 96-well plate (>90% viability; 1 ×10^4^ cells/well; 200 µL per well) were washed twice with phosphate-buffer saline (PBS), resuspended in PBS containing 0.4 μM DHR-123, and incubated for 10 min at 30 °C in the dark. Lastly, the fluorescence signal (λ_ex_ = 505 nm, λ_em_ = 535 nm) was recorded on a VersaFluor Fluorimeter (Bio-Rad, Marnes-la-Coquette, France).

#### 3.8.3. Measurement of Mitochondrial Membrane Potential (MMP)

The MMP (ΔΨ_m_) of HepG2 cells treated with green synthesized monometallic and bimetallic NPs was taken using 3,3′-dihexyloxacarbocyanine iodide (Sigma), which works by staining mitochondria according to their MMP. Cells were incubated in culture media supplied with 25 nM 3,3′-dihexyloxacarbocyanine iodide at 37 °C for 40 min. The MMP was expressed as relative fluorescent units (RFU). The experiment was performed thrice.

#### 3.8.4. Caspase-3 Gene Expression and Caspase-3/7 Activity

For the measurement of caspase-3 gene expression, total RNA was isolated and quantified by using an GeneJET RNA Purification Kit (Thermo Scientific, Sigma Aldrich, Waltham, MA, USA) and Quant-iT RNA Assay Kit (Invitrogen, Sigma Aldrich, Waltham, MA, USA), respectively. A first-strand cDNA synthesis kit (Thermo, Sigma Aldrich, Waltham, MA, USA) was used to perform reverse transcription. The PikoReal real-time PCR system (Thermo Fisher, Sigma Aldrich, Waltham, MA, USA) was used to perform quantitative PCR with the DyNAmo Color Flash SYBR Green qPCR Kit (Thermo Fisher). A 10 µL PCR reaction mixture was prepared which contained 0.5 µL diluted cDNAs, 2 × SYBR Green mix, and 1 µM of each primer pairs. The PCR reaction was carried out as follows: 7 min at 95 °C, 40 cycles of 10 s at 95 °C, 10 s at 60 °C, and 30 s at 72 °C. Data were analyzed using Pikoreal software. A total of 3 biological replicates and 2 technical repetitions were performed for each sample. The caspase-3 primers used were: 5′-TGTTTGTGTGCTTCTGAGCC-3′ (Forward primer) and 5′-CACGCCATGTCATCATCAAC-3′ (Reverse primer) (amplicon size: 210 bp).

Caspase-3/7 activity was measured by using frozen cells to obtain cytosolic protein extracts. A glass chilled mortar and pestle were used to ground the samples in 500 mL ice-cold extraction buffer containing 10% (*w*/*v*) sucrose, 100 mM HEPES (pH 7.2), 5 mM DTT, 1% (*v*/*v*) NP40, and 0.1% (*w*/*v*) CHAPS. Incubation of the homogenate was carried out on ice for 15 min, and it was centrifuged twice at 13,000× *g* for 10 min at 4 °C to pellet out cell debris. The supernatant was filtered through a 0.22 mm filter. Ab Apo-ONE Homogeneous Caspase-3/7 Assay kit (Promega, Sigma Aldrich, Madison, WI, USA) was used to measure in vitro caspase-3/7 activity, following the instructions of manufacturer.

### 3.9. Statistical Data Analysis 

All the data were evaluated statistically to determine the average values and the standard deviation using SPSS (Windows Version 7.5.1, SPSS Inc., Chicago, IL, USA). The data are expressed as the mean ± SE.

## 4. Conclusions

In this research, monometallic AgNPs, ZnONPs, and bimetallic Ag–ZnONPs were green synthesized from *Morus macroura* leaf extract under the influence of UV-C light for the first time. The phytochemical profile of *M. macroura* showed that the plant is enriched with phenolic and flavonoid contents, showing increased antioxidant activity. FTIR spectra showed the involvement of polyphenolic and carboxylic acid as well as aromatic compounds in the bio-reduction and stabilization of monometallic and bimetallic NPs. Characterization techniques such as SEM-EDX and XRD helped to draw the conclusion that UV-C light had a more profound effect on modeling the morphological features of green synthesized NPs. A reduction in size and more defined shapes were observed in the case of UV-C mediated green synthesis of NPs. The physico-chemical properties of NPs also determine their applications and influence their efficacy. In our study, UV-C mediated AgNPs showed exceptionally strong anti-diabetic and anti-aging activities. All NPs were moderately hemolytic towards hRBCs and moderately toxic towards brine shrimp, hence showing their bio-safe nature. Here, the role of UV-C irradiation in the green synthesis of monometallic and bimetallic NPs and ultimately in enhancing their in vitro biological activities has been described for the first time. Interestingly, the differences observed in biological activity between the control and UV-irradiated NPs could be attributable to differences in cellular absorption. Future research will be needed to test this theory. Once the process is further optimized and scaled up, these biocompatible NPs could be used as strong contenders in anti-cancerous and anti-diabetic therapeutics as well as in cosmetics.

## Figures and Tables

**Figure 1 ijms-22-11294-f001:**
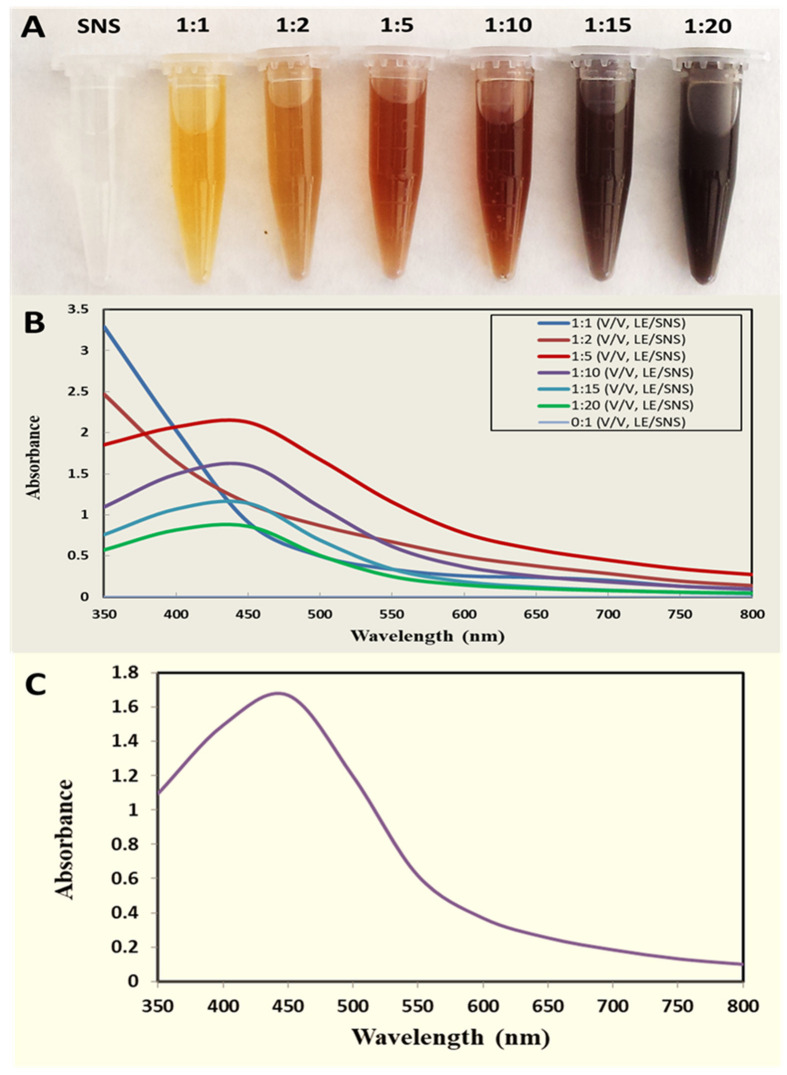
(**A**) Color change observed in reaction mixtures of AgNO_3_ solution (SNS) + leaf extract (LE) at different ratios (*v*/*v*). (**B**) Optimization spectra of AgNPs at different ratios as observed by a UV-visible spectrophotometer. (**C**) Evaluation of the stability of synthesized AgNPs using a UV-visible spectrophotometer after 1 month of incubation of reaction mixture.

**Figure 2 ijms-22-11294-f002:**
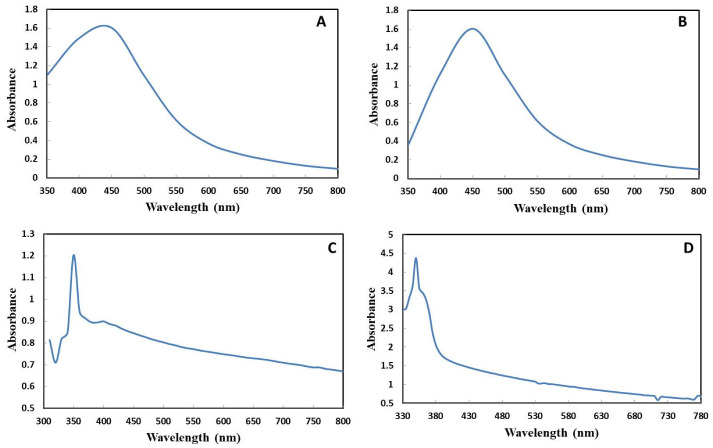
UV-visible spectra. (**A**) Control AgNPs. (**B**) UV-C mediated AgNPs. (**C**) Control ZnONPs. (**D**) UV-C mediated ZnONPs.

**Figure 3 ijms-22-11294-f003:**
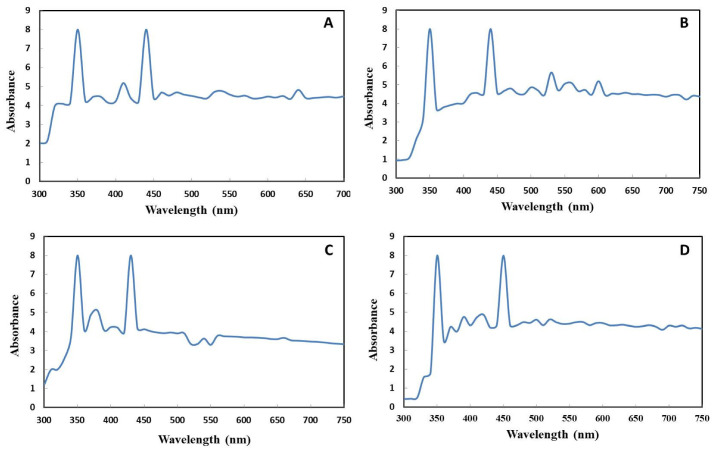
UV–visible spectra. (**A**) Control bimetallic nanoparticles of Ag–ZnO (0.1/0.1). (**B**) UV-C mediated bimetallic nanoparticles of Ag–ZnO (0.1/0.1). (**C**) Control bimetallic nanoparticles of Ag–ZnO (0.1/0.5). (**D**) UV-C mediated bimetallic nanoparticles of Ag–ZnO (0.1/0.5).

**Figure 4 ijms-22-11294-f004:**
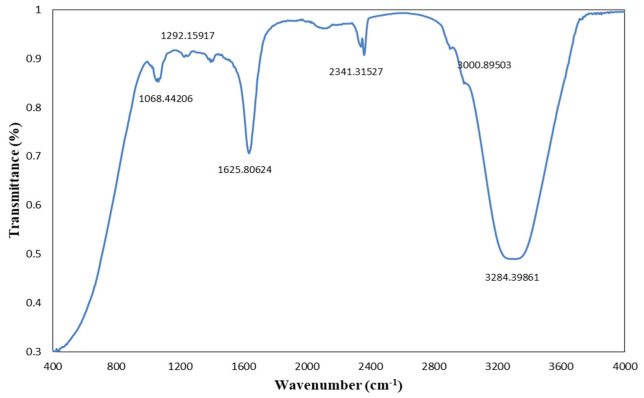
FTIR spectra of *M. macroura* leaf extract.

**Figure 5 ijms-22-11294-f005:**
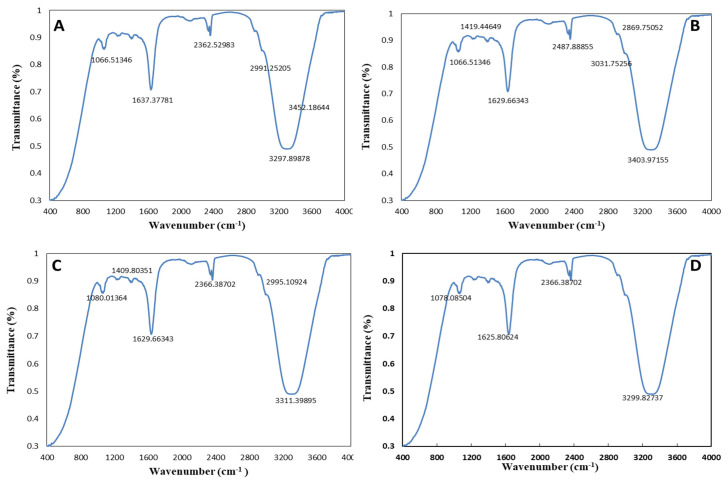
FTIR spectra of monometallic NPs. (**A**) Control AgNPs. (**B**) UV-C mediated AgNPs. (**C**) Control ZnONPs. (**D**) UV-C mediated ZnONPs.

**Figure 6 ijms-22-11294-f006:**
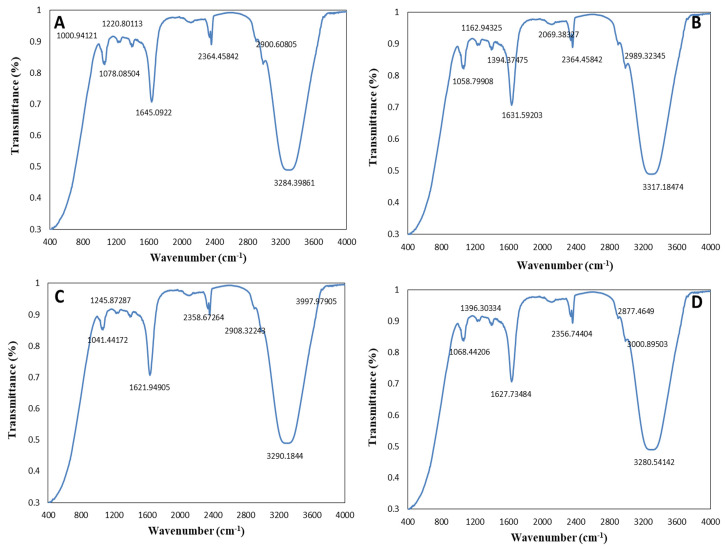
FTIR spectra of bimetallic Ag–ZnONPs. (**A**) Control Ag–ZnO (0.1/0.1). (**B**) UV-C mediated Ag–ZnO (0.1/0.1). (**C**) Control Ag–ZnO (0.1/0.5). (**D**) UV-C mediated Ag–ZnO (0.1/0.5).

**Figure 7 ijms-22-11294-f007:**
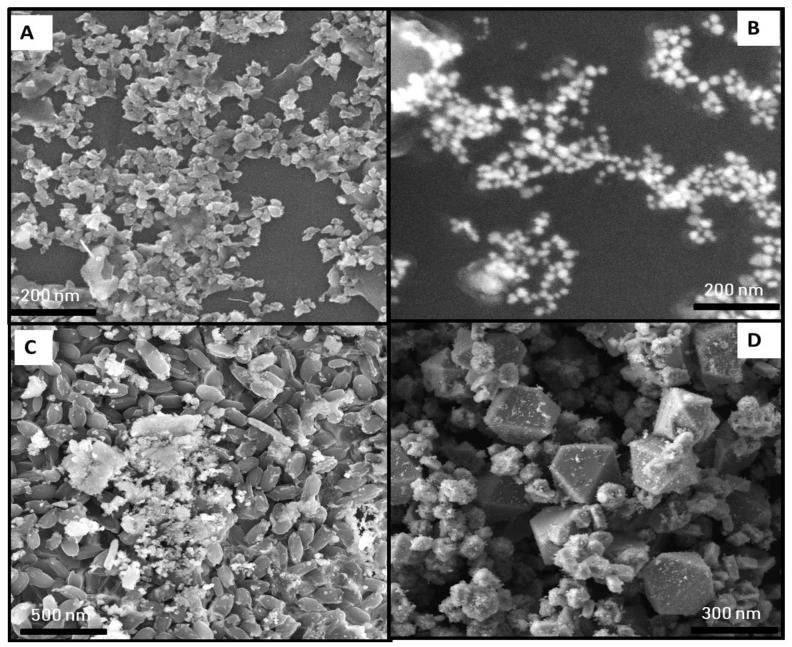
SEM images of monometallic NPs. (**A**) Control AgNPs. (**B**) UV-C mediated AgNPs. (**C**) Control ZnONPs. (**D**) UV-C mediated ZnONPs.

**Figure 8 ijms-22-11294-f008:**
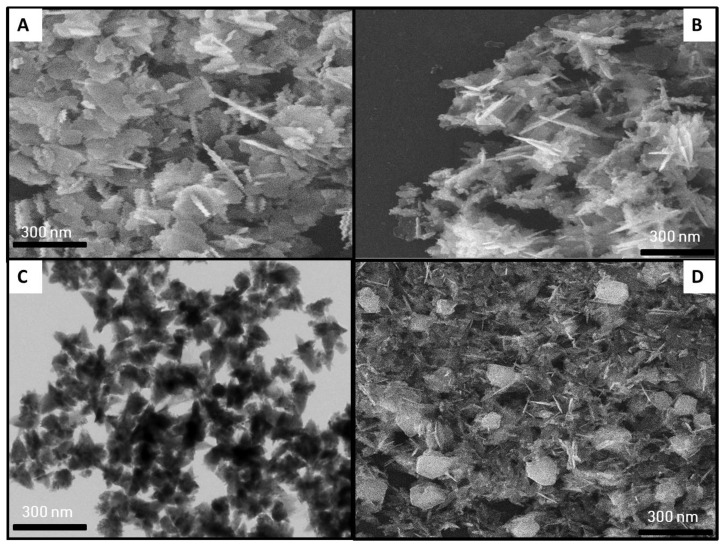
SEM images of bimetallic Ag–ZnONPs. (**A**) Control Ag–ZnO (0.1/0.1). (**B**) UV-C mediated Ag–ZnO (0.1/0.1). (**C**) Control Ag–ZnO (0.1/0.5). (**D**) UV-C mediated Ag–ZnO (0.1/0.5).

**Figure 9 ijms-22-11294-f009:**
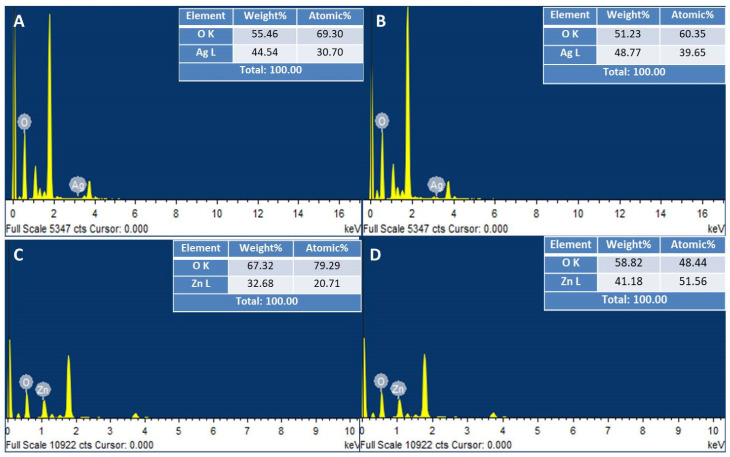
EDX analysis of monometallic NPs. (**A**) Control AgNPs. (**B**) UV-C mediated AgNPs. (**C**) Control ZnONPs. (**D**) UV-C mediated ZnONPs.

**Figure 10 ijms-22-11294-f010:**
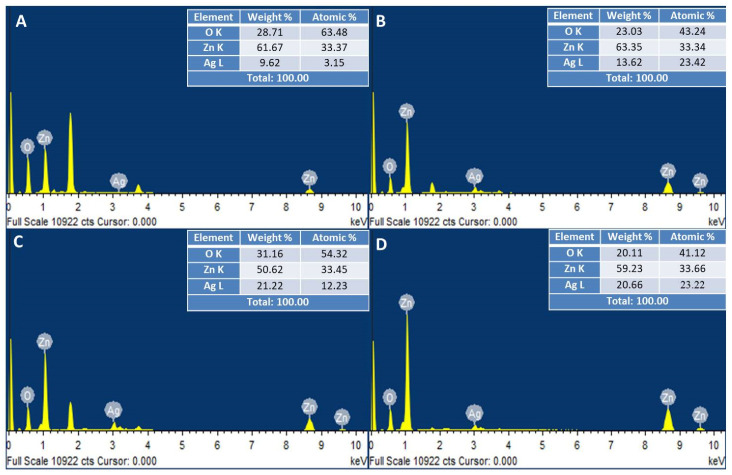
EDX analysis of bimetallic Ag–ZnONPs. (**A**) Control Ag–ZnO (0.1/0.1). (**B**) UV-C mediated Ag–ZnO (0.1/0.1). (**C**) Control Ag–ZnO (0.1/0.5). (**D**) UV-C mediated Ag–ZnO (0.1/0.5).

**Figure 11 ijms-22-11294-f011:**
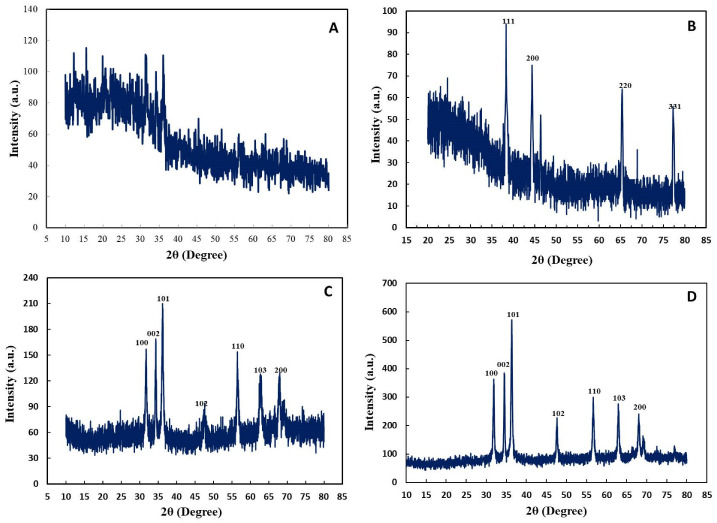
XRD pattern of monometallic NPs. (**A**) Control AgNPs. (**B**) UV-C mediated AgNPs. (**C**) Control ZnONPs. (**D**) UV-C mediated ZnONPs.

**Figure 12 ijms-22-11294-f012:**
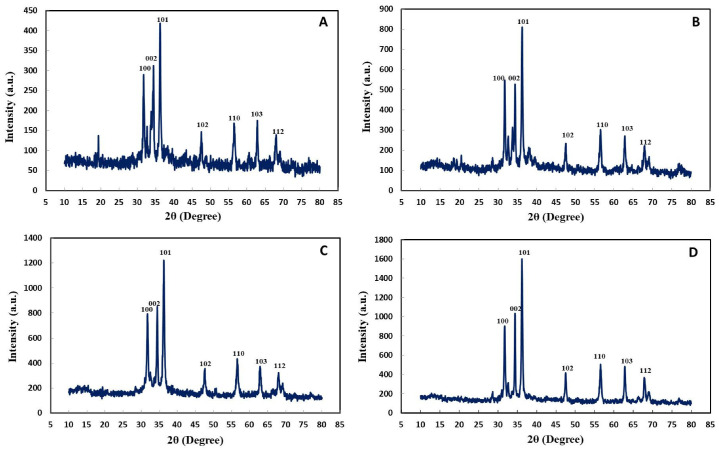
XRD pattern of bimetallic Ag–ZnONPs. (**A**) Control Ag–ZnO (0.1/0.1). (**B**) UV-C mediated Ag–ZnO (0.1/0.1). (**C**) Control Ag–ZnO (0.1/0.5). (**D**) UV-C mediated Ag–ZnO (0.1/0.5).

**Figure 13 ijms-22-11294-f013:**
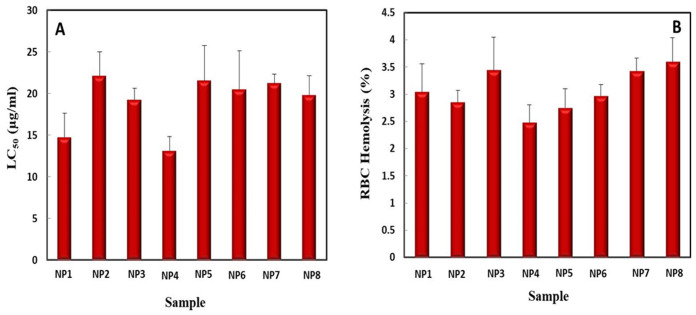
Biocompatibility studies of NPs. (**A**) Brine shrimp lethality assay of NPs. (**B**) Biocompatibility of NPs assessed with hRBCs. The data represent the mean of 3 replicates ± SE. (Note—NP1: control AgNPs; NP2: UV-C mediated AgNPs; NP3: control ZnONPs; NP4: UV-C mediated ZnONPs; NP5: control bimetallic Ag–ZnONPs (0.1/0.1); NP6: UV-C mediated bimetallic Ag–ZnONPs (0.1/0.1); NP7: control bimetallic Ag–ZnONPs (0.1/0.5); NP8: UV-C mediated bimetallic Ag–ZnONPs (0.1/0.5)).

**Figure 14 ijms-22-11294-f014:**
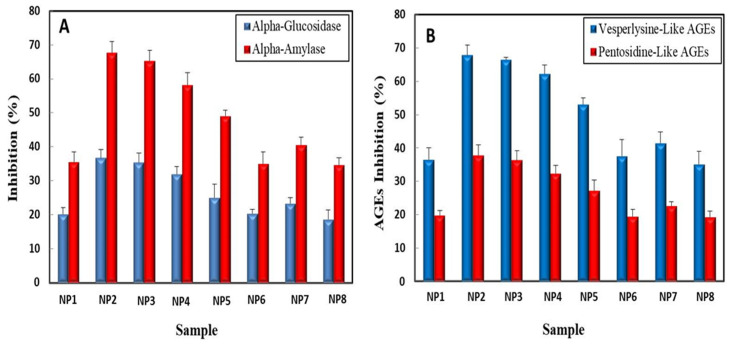
(**A**) Anti-diabetic activities of monometallic and bimetallic NPs assessed through the inhibition of α-glucosidase and α-amylase. (**B**) Anti-glycation activity of monometallic and bimetallic NPs measured through the inhibition of AGE production. Data represent the mean of 3 replicates ± SE. (Note—NP1: control AgNPs; NP2: UV-C mediated AgNPs; NP3: control ZnONPs; NP4: UV-C mediated ZnONPs; NP5: control bimetallic Ag–ZnONPs (0.1/0.1); NP6: UV-C mediated bimetallic Ag–ZnONPs (0.1/0.1); NP7: control bimetallic Ag–ZnONPs (0.1/0.5); NP8: UV-C mediated bimetallic Ag–ZnONPs (0.1/0.5)).

**Figure 15 ijms-22-11294-f015:**
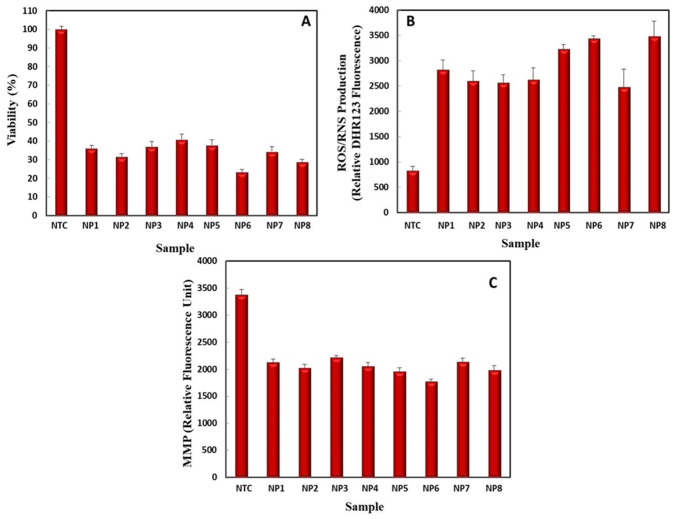
Anti-cancerous activities of monometallic and mimetallic NPs measured in terms of (**A**) cell viability, (**B**) intracellular ROS/RNS production, and (**C**) mitochondrial membrane potential disruption. Data represent the mean of 3 replicates ± SE. (Note—NTC: non-treated cells; NP1: control AgNPs; NP2: UV-C mediated AgNPs; NP3: control ZnONPs; NP4: UV-C mediated ZnONPs; NP5: control bimetallic Ag–ZnONPs (0.1/0.1); NP6: UV-C mediated bimetallic Ag–ZnONPs (0.1/0.1); NP7: control bimetallic Ag–ZnONPs (0.1/0.5); NP8: UV-C mediated bimetallic Ag–ZnONPs (0.1/0.5)).

**Figure 16 ijms-22-11294-f016:**
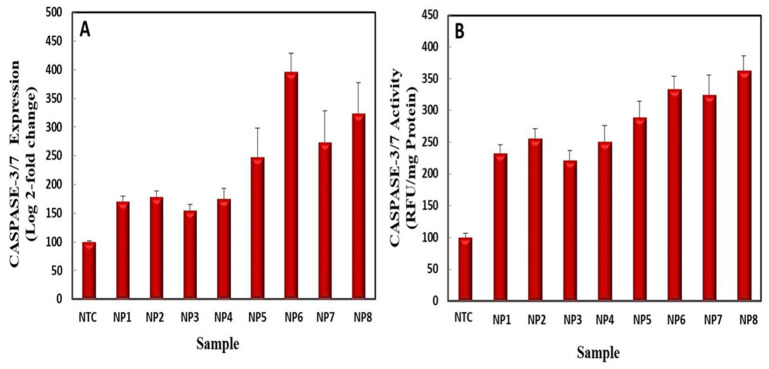
(**A**) Caspase-3 gene expression in HepG2 cell lines. (**B**) Caspase-3/7 activity in HepG2 cells when exposed to monometallic and bimetallic NPs. Data represent the mean of 3 replicates ± SE. (Note—NTC: non-treated cells; NP1: control AgNPs; NP2: UV-C mediated AgNPs; NP3: control ZnONPs; NP4: UV-C mediated ZnONPs; NP5: control bimetallic Ag–ZnONPs (0.1/0.1); NP6: UV-C mediated bimetallic Ag–ZnONPs (0.1/0.1); NP7: control bimetallic Ag–ZnONPs (0.1/0.5); NP8: UV-C mediated bimetallic Ag–ZnONPs (0.1/0.5)).

**Table 1 ijms-22-11294-t001:** The average diameter of nanoparticles calculated using XRD data.

Nanoparticles Type	Average Diameter (nm)
**Control AgNPs**	147.23
**UV-C mediated AgNPs**	22.93
**Control ZnONPs**	53.21
**UV-C mediated ZnONPs**	37.03
**Control Ag–ZnONPs (0.1/0.1)**	24. 21
**UV-C mediated Ag–ZnONPs (0.1/0.1)**	21.69
**Control Ag–ZnONPs (0.1/0.5)**	40.11
**UV-C mediated Ag–ZnONPs (0.1/0.5)**	23.65

## Data Availability

All data are included in the present study.

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
