# Peer review of "Light Tailoring: Impact of UV-C Irradiation on Biosynthesis, Physiognomies, and Clinical Activities of Morus macroura-Mediated Monometallic (Ag and ZnO) and Bimetallic (Ag–ZnO) Nanoparticles"

_ijms, 2021, doi:10.3390/ijms222011294_

Round 1

Reviewer 1 Report

  1. This study designs well and presents some results that are valuable for the related research field. Green nanotechnology is a promising field for a number of biomedical applications.
  2. Add significant difference between means in all data.
  3. Add scale bars in figure 8.
  4. line 23: During synthesis of nanoparticles - During the synthesis of nanoparticles
  5. line 36: In case - In the case; Also make the collection throughout the manuscript.
  6. line 51:  interest of scientists in NPs research is intense - the interest of scientists in NPs research is intense
  7. line 54: electronic - electronics
  8. line 57: but on the other hand lead to increase use of toxic chemicals - but on the other hand, lead to an increase in the use of toxic chemicals
  9. line 62: from green approach - from the green approach
  10. line 276: to previously described protocol - to the previously described protocol
  11. line 529: Furthermore, increase in Zinc acetate molarity - Furthermore, an increase in Zinc acetate molarity
  12. line 696: displayed highest inhibition - displayed the highest inhibition
  13. line 744: overwhelmed the anti-oxidant capacity of cell, eventually leading to death - overwhelmed the antioxidant capacity of cells, eventually leading to the death
  14. line 833: Phytochemical profile of M. Macroura - The phytochemical profile of M. Macroura
  15. line 838: in modelling - in modeling
  16. The layout and resolution of figures 13, 14, 15, 16, need to be improved significantly.

Author Response

Reviewer-1

Comments: This study design well and presents some results that are valuable for the related research field. Green nanotechnology is a promising field for a number of biomedical applications.

AUTHORS: Thank you very much for your comments and suggestions that greatly help us to improve the quality of the present revised version of our MS. We do our best to answer to all your queries and hope this revised version will answer them. Revisions appear in track changes.

  1. Add significant difference between means in all data.

AUTHORS: Thank you very much for your valuable suggestion. Significant difference between means in all the data in text has been added.

  1. Add scale bars in Figure 8.

AUTHORS: Scale bars in Figure 8 has been added.

  1. Line 23: During synthesis of nanoparticles – During the synthesis of nanoparticles.

AUTHORS: Thank you for pointing out this mistake. Line has been corrected accordingly.

  1. Line 36: In case – In the case; Also make collection throughout the manuscript.

AUTHORS: Thank you for pointing out this mistake. Lines has been corrected accordingly throughout the manuscript.

  1. Line 51: interest of scientists in NPs research is intense – the interest of scientists in NPs research is intense.

AUTHORS: Thank you. Line has been corrected accordingly.

  1. Line 54: electronic – electronics

AUTHORS: Thank you. Line has been corrected accordingly.

  1. Line 57: but on the other hand lead to increase use of toxic chemicals - but on the other hand, lead to an increase in the use of toxic chemicals.

AUTHORS: Thank you. Line has been corrected accordingly.

  1. Line 62: from green approach - from the green approach

AUTHORS: Thank you. Line has been corrected accordingly.

  1. Line 276: to previously described protocol - to the previously described protocol.

AUTHORS: Thank you. Line has been corrected accordingly.

  1. Line 529: Furthermore, increase in Zinc acetate molarity - Furthermore, an increase in Zinc acetate molarity.

AUTHORS: Thank you. Line has been corrected accordingly.

  1. Line 276: to previously described protocol - to the previously described protocol.

AUTHORS: Thank you. Line has been corrected accordingly.

  1. Line 696: displayed highest inhibition - displayed the highest inhibition.

AUTHORS: Thank you. Line has been corrected accordingly.

  1. Line 744: overwhelmed the anti-oxidant capacity of cell, eventually leading to death - overwhelmed the antioxidant capacity of cells, eventually leading to the death

AUTHORS: Thank you. Line has been corrected accordingly.

  1. Line 833: Phytochemical profile of M. Macroura - The phytochemical profile of M. Macroura.

AUTHORS: Thank you. Line has been corrected accordingly.

  1. Line 838: in modelling - in modeling.

AUTHORS: Thank you. Line has been corrected accordingly.

  1. The layout and resolution of figures 13, 14, 15, 16, need to be improved significantly.

AUTHORS: Thank you for your valuable suggestion. Layouts of all figures have been improved.

Reviewer 2 Report

This article reports on the green synthesis of AgNPs, ZnONPs and their bimetallic nanoparticles by using the leaf extract of the plant Morus Macroura. The syntheses were conducted with and without UV-C irradiation and the difference in nanoparticle characteristics and biological activities of irradiated and non-irradiated nanoparticles was show. The synthesis is original and the biological effects of the as-synthesized nanoparticles are very interesting. However, some questions are unanswered.

1) Line 364-367 and line 372-373: For a better readability, it would be great when your measurements and the values cited from literature have the same unit of measurement.

2) Figure 1: Could you please explain the abbreviations LE and SNS in the figure caption. Also, you mentioned in line 398 that you measured the absorbance of the AgNP solutions after 1 month. Could you please show the absorption spectrum?

3) You explained the differences in the absorption spectrum of non-irradiated and irradiated nanoparticle solution with the difference in size distribution and agglomeration. Therefore, DLS (dynamic light scattering) measurements of the nanoparticle solution should be performed to prove the differences in size distribution.

4) Line 513: The size UV-C irradiated ZnONPs (57.23-62,98 nm) is smaller than the control ZnONPs (69.23-75.23 nm). However, the scale bar of Figure 7D (irradiated ZnONPs) with 10 µm is much larger than the scale bar of figure 7C (control ZnONPs) with 500 nm. Is the scale bar of figure 7D correct? Then, the irradiated ZnONPs are much larger than the non-irradiated ones.

5) Figure 8: Please insert the scale bars.

6) Line 525: You stated that the AgNPs are incorporated into the ZnOPs lattice. How do you recognize this in the SEM image? Line 530: You describe the nanoparticles in Figure 8C as flower (ZnO) and hexagonal (Ag). It is hard to distinguish between ZnO and Ag in the SEM images. For this reason, it would be beneficial to show the images of the STEM-EDX elemental mapping to see the distribution of the Zn and Ag atoms. (For example: S. Klein et al. ACS Appl. Bio Mater. 2018, doi: 10.1021/acsabm.8b00511).

7) Line 613: You mentioned that the sizes calculated from the XRD spectra and measured by SEM are similar. Could you please show the results?

8) You show the reduction in the cell viability of HepG2, reduction in MMP and increase in ROS production of the synthesized nanoparticles. The nanoparticles show a great anti-cancer potential. How is the effect of these nanoparticles on non-tumorigenic cell lines? To emphasize the anti-cancer potential, it would be great to show that the nanoparticles do not have the same effect on a non-tumorigenic cell line.

9) You mentioned that there is a difference in the biological activity of control and UV-irradiated nanoparticles. This has also an influence on the size and shape of these nanoparticles. Could this difference be due to a difference in the cellular uptake of these nanoparticles? Could you show some TEM images with internalized nanoparticles? Or could you prove if there is a difference in cellular uptake (for example by measuring the intracellular Ag or Zn content)?

Minor corrections:

  • Line 26: [(0.1/0.5) and 0.1/05.5)] should be [(0.1/0.1) and (0.1/0.5)]
  • Line 283: α-.Amylase … α-Amylase
  • Line 296: shah et al. …. Shah et al.
  • Line 315: incubated of 3h …. Incubated for 3h

Author Response

Reviewer-2

Comments: This article reports on the green synthesis of AgNPs, ZnONPs and their bimetallic nanoparticles by using the leaf extract of the plant Morus Macroura. The syntheses were conducted with and without UV-C irradiation and the difference in nanoparticle characteristics and biological activities of irradiated and non-irradiated nanoparticles was show. The synthesis is original and the biological effects of the as-synthesized nanoparticles are very interesting. However, some questions are unanswered.

AUTHORS: Dear Sir, Thank you very much for your comments and suggestions that greatly help us to improve the quality of our MS. We do our best to answer to all your queries and hope this revised version will answer them. All revisions appear in track changes.

  1. Line 364-367 and line 372-373: For a better readability, it would be great when your measurements and the values cited from literature have the same unit of measurement.

AUTHORS:  Sir, we agree with your suggestion. But, we did not find any report published in literature in which the total phenolic or flavonoid contents of Morus Macroura plant extract were measured in the same unit as we reported and to strengthen our results, it was necessary to compare data of the same plant published previously. Therefore, we compare with the available data. Rest of the results values cited from literature have the same units of measurement.

  1. Figure 1: Could you please explain the abbreviations LE and SNS in the figure caption. Also, you mentioned in line 398 that you measured the absorbance of the AgNPs solutions after 1 month. Could you please show the absorption spectrum?

AUTHORS: We have provided the abbreviations of LE (leaf extract) and SNS (silver nitrate solution) in the figure 1 and also provided the absorption spectrum of AgNPs after one month in the Figure 1.

  1. You explained the differences in the absorption spectrum of non-irradiated and irradiated nanoparticle solution with the difference in size distribution and agglomeration. Therefore, DLS (dynamic light scattering) measurements of the nanoparticle solution should be performed to prove the differences in size distribution.

AUTHORS: Dear Sir, Thank you very much for suggestion. As this statement is given on the basis of previous published literature that the UV-Vis absorption peaks of nanoparticles can give a clue about the sizes of nanoparticles by observing the broadness or narrowness of the SPR peaks. So, we hypothesized that the difference in SPR peaks of nanoparticles could be attributed to difference in size distribution. However, we have modified this statement in revised manuscript and currently we have not access to DLS, therefore could not perform this analysis.

  1. Line 513: The size UV-C irradiated ZnONPs (57.23-62.98 nm) is smaller than the control ZnONPs (69.23-75.23 nm). However, the scale bar of Figure 7D (irradiated ZnONPs) with 10 µm is much larger than the scale bar of figure 7C (control ZnONPs) with 500 nm. Is the scale bar of figure 7D correct? Then, the irradiated ZnONPs are much larger than the non-irradiated ones.

AUTHORS: Thank you very much for pointing out this. The scale bar of figure 7D was mistakenly written wrong. We have corrected in figure 7D in revised manuscript.

  1. Figure 8: Please insert the scale bars.

AUTHORS: We have inserted the scale bars in figure 8A-D.       

  1. Line 525: You stated that the AgNPs are incorporated into the ZnONPs lattice. How do you recognize this in the SEM image? Line 530: You describe the nanoparticles in Figure 8C as flower (ZnO) and hexagonal (Ag). It is hard to distinguish between ZnO and Ag in the SEM images. For this reason, it would be beneficial to show the images of the STEM-EDX elemental mapping to see the distribution of the Zn and Ag atoms. (For example: S. Klein et al. ACS Appl. Bio Mater. 2018, doi: 10.1021/acsabm.8b00511).

AUTHORS: In Line 525 and 530, it was hypothesized by observing the SEM images, but we agree with you it was really hard to find any significant difference between Ag and ZnO nanoparticles morphology. Therefore, we have modified both statement in the revised manuscript according to your suggestions. Moreover, we have currently no access to STEM-EDX elemental mapping due to covid situation, and moreover this was also not the objective of our study.

  1. Line 613: You mentioned that the sizes calculated from the XRD spectra and measured by SEM are similar. Could you please show the results?

AUTHORS: The average sizes of the nanoparticles calculated by using XRD data are shown in table 1 in revised manuscript.

  1. You show the reduction in the cell viability of HepG2, reduction in MMP and increase in ROS production of the synthesized nanoparticles. The nanoparticles show a great anti-cancer potential. How is the effect of these nanoparticles on non-tumorigenic cell lines? To emphasize the anti-cancer potential, it would be great to show that the nanoparticles do not have the same effect on a non-tumorigenic cell line.

AUTHORS: The toxicity and biocompatibility have been checked using two different models: brine shrimp lethality Assay as well as non-tumorigenic human red blood cells (Figure 13). Both assays showed no potential toxicity and evidenced the biocompatibility with non-tumorigenic human cells.

  1. You mentioned that there is a difference in the biological activity of control and UV-irradiated nanoparticles. This has also an influence on the size and shape of these nanoparticles. Could this difference be due to a difference in the cellular uptake of these nanoparticles? Could you show some TEM images with internalized nanoparticles? Or could you prove if there is a difference in cellular uptake (for example by measuring the intracellular Ag or Zn content)?

AUTHORS: Thank you very much for pointing this hypothesis. We have included this hypothesis in the revised version of the manuscript. This is a really interesting point that deserves to be explored more in the future. Not all hypotheses are supposed to be tested. Future research will be conducted in cooperation with teams capable of doing such relevant and reproducible imaging analyses in order to investigate this possible difference in internalization. We are unable to provide an analysis worthy of publishing at this time due to our present level of expertise in the field of imaging.

  1. Line 26: [(0.1/0.5) and 0.1/05.5)] should be [(0.1/0.1) and (0.1/0.5)]

AUTHORS: Line has been corrected accordingly.

  1. Line 283: α-.Amylase … α-Amylase

AUTHORS: Line has been corrected accordingly.

  1. Line 296: shah et al. …. Shah et al.

AUTHORS: Line has been corrected accordingly.

  1. Line 315: incubated of 3h …. Incubated for 3h

AUTHORS: Line has been corrected accordingly.

Round 2

Reviewer 2 Report

All my suggestions and questions are attended. Therefore, I recommend to accept the manuscript in the present form.